# Cholesterol esters form supercooled lipid droplets whose nucleation is facilitated by triacylglycerols

Calvin Dumesnil[1,6], Lauri Vanharanta [2,3,6], Xavier Prasanna [4,6], Mohyeddine Omrane[1], Maxime Carpentier[1], Apoorva Bhapkar[1], Giray Enkavi [4], Veijo T. Salo[2,3,5], Ilpo Vattulainen [4] ✉, Elina Ikonen [2,3] ✉ & Abdou Rachid Thiam[1] ✉

Cellular cholesterol can be metabolized to its fatty acid esters, cholesteryl esters (CEs), to be stored in lipid droplets (LDs). With triacylglycerols (TGs), CEs represent the main neutral lipids in LDs. However, while TG melts at ~4 °C, CE melts at ~44 °C, raising the question of how CE-rich LDs form in cells. Here, we show that CE forms supercooled droplets when the CE concentration in LDs is above 20% to TG and, in particular, liquid-crystalline phases when the fraction of CEs is above 90% at 37 °C. In model bilayers, CEs condense and nucleate droplets when the CE/phospholipid ratio reaches over 10-15%. This concentration is reduced by TG pre-clusters in the membrane that thereby facilitate CE nucleation. Accordingly, blocking TG synthesis in cells is sufficient to strongly dampen CE LD nucleation. Finally, CE LDs emerged at seipins, which cluster and nucleate TG LDs in the ER. However, when TG synthesis is inhibited, similar numbers of LDs are generated in the presence and absence of seipin, suggesting that seipin controls CE LD formation via its TG clustering capacity. Our data point to a unique model whereby TG pre-clusters, favorable at seipins, catalyze the nucleation of CE LDs.

From plant to human, dozens of neutral lipids can be made by cells and deposited into lipid droplets (LDs)[1]. LDs control cellular energy and lipid homeostasis but also possess several other functions[2] attuned to the LD formation triggering cue[2–4]. LD formation is initiated when sufficient amounts of neutral lipids accumulate in the endoplasmic reticulum (ER)[5,6]. As neutral lipids are often apolar and hydrophobic, they are released in the hydrophobic core of the ER bilayer[1]. There, neutral lipids condense and nucleate a nascent LD which grows by acquiring more neutral lipids and buds off as a spherical LD out of the ER bilayer[7]. Despite many advances in understanding LD biogenesis[8–14], the molecular mechanisms underlying the different steps of LD formation are not fully resolved.

Triacylglycerols (TGs) are the major neutral lipids in mammalian cells and, in silico, they can condense and nucleate a droplet in a bilayer if their ratio to phospholipids exceeds 3–4%[15–17]. In cells, this concentration is altered by several lipid and protein factors, including seipin[5,8]. Seipin is an integral ER protein, forming an oligomeric donut shape[10,18–20] and displaying favorable interactions with TG. Seipin decreases the TG nucleation concentration to ~1.25%[10,13,14] to induce nascent LD formation and growth[21–23]. In seipin deletion, more TG LDs

[1]Laboratoire de Physique de l'École normale supérieure, ENS, Université PSL, CNRS, Sorbonne Université, Université Paris Cité, F-75005 Paris, France. [2]Department of Anatomy and Stem Cells and Metabolism Research Program, Faculty of Medicine, University of Helsinki, Helsinki, Finland. [3]Minerva Foundation Institute for Medical Research, Helsinki, Finland. [4]Department of Physics, University of Helsinki, Helsinki, Finland. [5]Structural and Computational Biology Unit, European Molecular Biology Laboratory (EMBL), Heidelberg, Germany. [6]These authors contributed equally: Calvin Dumesnil, Lauri Vanharanta, Xavier Prasanna. ✉e-mail: ilpo.vattulainen@helsinki.fi; elina.ikonen@helsinki.fi; thiam@ens.fr

are nucleated[11,24,25] but they fail to grow normally[12]. Thereupon, seipin ensures that only a few TG LDs are nucleated and mature properly[1,12].

Cells can make other neutral lipids such as squalene, an intermediate in the cholesterol biosynthesis pathway. Squalene alone fails to form LDs in yeast and accumulates in the ER and in vitro membranes[16,26,27]. Therefore, squalene seems to have a higher nucleation concentration than TG. Retinyl esters represent another neutral lipid class found in liver stellate cells. In yeast mutant cells lacking neutral lipids, the biosynthesis of retinyl palmitate, induced by the overexpression of the lecithin retinol acyltransferase, leads to the formation of retinyl palmitate LDs[28], which can form away from seipin's location[28]. Seipin may therefore preferentially act on some neutral lipids[1,16]. How exactly neutral lipids impact LD formation and whether/how seipin is involved in the nucleation of LDs composed of neutral lipids other than TG remain unknown.

Together with TGs, cholesterol esters (CEs) represent the most abundant neutral lipid in mammalian cells. TG results from diacylglycerol esterification and CE from cholesterol esterification to a fatty acid. Depending on the cell type, TG/CE ratios vary significantly. For instance, white adipocytes, specialists in long-term energy storage, can make an ultra-large TG-rich LD, tens of μm in diameter. Macrophages can make CE-rich LDs and become foam cells during atherogenesis. In some cells, mixed TG/CE LDs are generated while in others, such as adrenocortical cells, specialized in steroid hormone synthesis, spatially distinct TG and CE LDs can form[29].

TGs and CEs are chemically highly divergent and therefore might require different membrane physical chemistry and protein settings to be packaged into LDs. For instance, triolein (TG) melts at -4 °C while cholesterol oleate (CE) only melts at -44 °C, and TG escapes curved membrane regions in contrast to CE[11,27]. Whether CE molecules can condense into a forming LD in the ER bilayer as TGs do and whether seipin mediates the formation of CE LDs similarly as it does for TG LDs, is unclear. We addressed these questions in this manuscript.

## Results

### CEs form supercooled liquid droplets

Since the CE melting point is -44 °C, we asked whether CE can be emulsified at 37 °C. We heated CE in a test tube to 37 °C and it remained solid as one would expect. At 50 °C, it formed a liquid phase but solidified when the temperature was brought back to 37 °C (Fig. 1A). At 37 °C, adding a buffer phase and mixing was insufficient to emulsify the CE powder (Supplementary Figure 1A). Thus, bulk CE cannot remain at equilibrium in a liquid state at 37 °C.

Next, we liquefied the CE sample and added the buffer at 50 °C. Vortexing the mixture allowed us to generate micrometer-sized droplets. Remarkably, when the emulsion was then cooled to 37 °C (or to 25 °C), the droplets stayed liquid (Fig. 1A). The droplets were liquid since they splashed at the water-air interface (Fig. 1B). Also, at the water-air surface, they displayed onion rings, features reminiscent of unstable liquids; for comparison, TG did not show such features (Supplementary Movies 1, 2). These observations indicated that the CE droplets were trapped in local minimum energy, below their melting point. Such a physical phenomenon is known as supercooling: trapping of liquids in metastable states between their liquid and stable solid states[30]. In solution, our generated CE droplets were short-lived, less than 72 h under rotation, as they fused, grew, and crystallized. Instead, when we generated the droplets in the presence of phospholipids to cover their interface with water, fusion, and crystallization were prevented and the lifetime of the droplets was prolonged for more than ten days (Supplementary Figure 1B).

The crystallization of supercooled liquids can be triggered by nucleation seeds[30]. Accordingly, when CE droplets met solid seeds at the air-water interface, they crystallized (Fig. 1C, Supplementary Movie 3). This observation confirms that the CE droplets were indeed supercooled and explains why we were not able to make ultra-large

millimetric CE droplets (in contrast to TG). Indeed, droplets of larger size crystallized more readily and had a higher probability to exit supercooling[30]. Finally, the stability of supercooled droplets depends on the nature of interfacial interactions[30]. By replacing the water phase with silicone oil, we lost the supercooled state, as droplets developed spikes at their interface and crystallized (Supplementary Figure 1C). This observation suggests that the interaction of CE with water molecules at the droplet interface contributed to the supercooled state of the droplets.

Since CE molecules may form anisotropic or liquid crystalline phases[31-34], we prepared CE droplets and imaged them under polarized light, which can reveal such organization. At 25 °C or 37 °C, we observed a mosaic of internal organizations of CEs in the droplets. The most frequent phenotype was droplets with Maltese crosses (Fig. 1D, E, Supplementary Figure 1D). Such a signal was indicative of a smectic liquid crystalline phase with an azimuthal organization of CE molecules[35], similar to the liquid crystalline lattices or "onion rings" seen by electron microscopy[33,34,36]. We also observed isotropic signals indicating no clear internal organization, but this phenotype was less frequent than the crystalline one (Fig. 1E). Other droplets had intermediate signals between liquid crystalline and isotropic signatures, or uninterpretable organization (Fig. 1D, E, Supplementary Figure 1D). Since the liquids are metastable, these variable organizations likely depend on the preparation method. In any case, pure CE droplets may have different internal organizations but the liquid crystalline state dominates.

Of note, the physical state of the CE droplets is defined by their free energy state which depends on several parameters such as temperature, pressure, and chemical potential (Supplementary Figure 1F). Here, we tuned the temperature to reach the energy of the supercooled state. Evidently, mammalian cells would not tune this parameter but rather modulate the chemical environment. In any case, regardless of the energetic path taken, our data show that pure CE droplets can be generated under a supercooled state, particularly facilitated by interfacial interactions with water (Supplementary Figure 1C). They are mainly trapped in the supercooled smectic liquid crystalline regime. As a consequence, the larger the CE droplet, the higher its propensity to crystallize.

### The appearance of the liquid crystalline phase in CE LDs depends on TG/CE ratio

We studied how the physical state of the CE droplets is impacted by TG. In bulk, CE was insoluble in TG (or diacylglycerol) above ~20% molar ratio at 37 °C (Supplementary Figure 1E, G). When droplets were directly made at 37 °C with CE/TG mixtures above 20%, we observed a de-mixed CE crystal and a TG liquid phase (Supplementary Figure 1E, G). Only when the sample was heated to liquefy the blend at 50 °C droplets could be made, as above, before the temperature was cooled to 37 °C. Consequently, CE/TG mixtures can form stable liquid droplets when CE is below ~20% and become supercooled above this ratio.

We then imaged the droplets under polarized light upon heating the sample from 30 °C to 37 °C. As expected, below 20% CE, the droplets were isotropic. Above 20%, they were also mostly isotropic, despite being supercooled, and displayed a smectic liquid crystalline phase above ~80% at 30 °C and ~93% at 37 °C (Fig. 1F, Supplementary Figure 1H). These results indicate that the TG/CE ratio tuned the physical state of the droplets and that the liquid crystalline regime is reached whenever CE goes above ~80–90%.

We next probed whether the CE/TG ratio can modulate the state of cellular LDs, by taking advantage of the liquid crystalline signature upon polarized light illumination. We employed A431 cells and loaded them with 200 μM cholesterol complexed with methyl-ß-cyclodextrin (MßCD) for 24 h to make CE-rich LDs, and then imaged the cells at 37 °C (Fig. 1G, Supplementary Figure 1I). We found 95% of the LDs to display a Maltese cross signal, the smectic liquid crystalline signature,

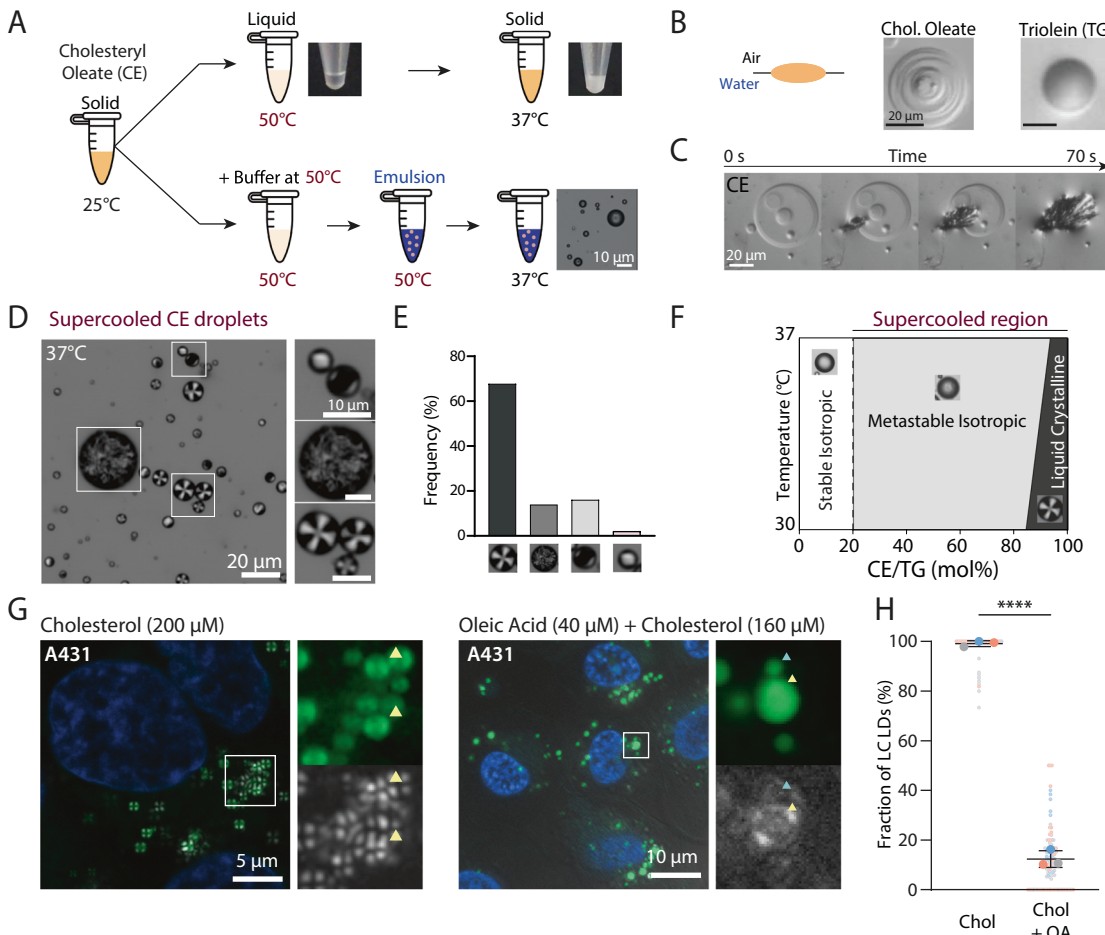

**Fig. 1 | CE can form supercooled droplets in vitro. A** Top: Schematic representation of bulk CE (here, cholesteryl oleate) upon heating from 25 °C to 50 °C and cooling down to 37 °C. Bulk CE liquefied when heated above its melting point and solidified upon cooling to 37 °C. Bottom: Schematic representation of CE emulsification protocol. CE was heated above its melting point and HKM buffer was added at 50 °C. The mixture was emulsified by vortexing and sonicating, and then imaged under confocal microscopy. The droplets appeared to be (meta)stable and thus supercooled. **B** CE and TG (here, triolein) emulsions were imaged at the water-air interface in brightfield. Left: Schematic representation of an oil droplet spreading at the water-air interface. Middle: image of a CE droplet spreading at the water-air interface. Some droplets revealed concentric rings. Right: Image of a TG droplet spreading. No concentric rings were observed. **C** Live imaging of the crystallization of a CE droplet upon meeting with a nucleation seed. **D** Supercooled CE droplets were imaged using polarized light microscopy on a temperature-controlled stage. Left: Image of a CE emulsion at 37 °C. Right: Images of the three polarized droplets phenotypes encountered. **E** Analysis of the occurrences of the lipid droplets phenotypes, $n = 236$ droplets. **F** Schematic segmentation of the CE/TG droplets' physical states according to the CE/TG ratio and emulsion temperature. Droplets are stable and isotropic below 20% of CE, metastable but isotropic until ~85% of CE near room temperature, and exhibit a liquid crystalline phase above. Done at 0%, 20%, 50%, 85%, 95%, and 100%. **G** A431 cells imaged after 24 h of cholesterol (200 μM) or oleic Acid (40 μM) +cholesterol (160 μM) feeding. Bodipy was added upon imaging for LD labeling and further analysis (**H**). Experiments done in triplicates. **H** Analysis of the fraction of LDs in LC form in **G** Mean ± SD. $n = 66$, 54, and 39 cells for 200 μM cholesterol. And 54, 25, and 27 cells for 160 μM cholesterol + 40 μM oleic acid. Each color point represents a data point from a replicate. The experiment was independently repeated three times with similar results. ****$p < 0.0001$ two-tailed Nested $t$ test.

suggesting that they were more than 90% enriched in CE. To vary the TG level, we supplemented the cells with both oleic acid and cholesterol at 20/80 (Fig. 1G, Supplementary Figure 1J). We found a significant decrease in the number of LDs in the liquid crystalline state (Fig. 1H, Supplementary Figure 1J, L, M). This result argues first that A431 cells generated mixed LDs containing both CEs and TGs and second, that in the presence of oleic acid, LDs had increased TG levels, which disrupted the liquid crystalline organization, in line with our in vitro experiment (when TG/CE levels are above ~1/9 at 37 °C; Fig. 1F). Accordingly, the liquid crystalline phase phenotype of LDs was restored in cells loaded with both oleic acid and cholesterol in the presence of pharmacological inhibitors of diacylglycerol O-acyltransferase 1 and 2 enzymes (DGAT 1, 2) that synthesize TG (Supplementary Figure 1K, L).

Together, these data indicate that the TG/CE ratio determines the molecular organization of LDs. Based on our in vitro studies, LDs would be under stable conditions when the CE concentration in LDs is below 20% relative to TG at 37 °C; at above 20% CE, LDs would be in a supercooled state and, particularly, in a smectic liquid crystalline phase at above 90% CE.

## Inhibition of TG synthesis compromises and stimulation of TG synthesis enhances CE LD formation in cells

As TG determined the physical states of CE-containing LDs, we asked whether it has a role in the biogenesis of such LDs. We, therefore, tested if inhibition of TG synthesis affects CE LD formation in A431 cells loaded with increasing amounts of cholesterol from MßCD for 1 h when LDs appeared and CE levels increased (Supplementary Figure 2A, B). This revealed that inhibition of DGAT 1 + 2 activity during cholesterol loading strongly compromised CE LD formation. At 50 μM cholesterol, while few LDs were generated in WT cells, almost none was made in the DGAT-inhibited condition (Fig. 2A, B). This observation

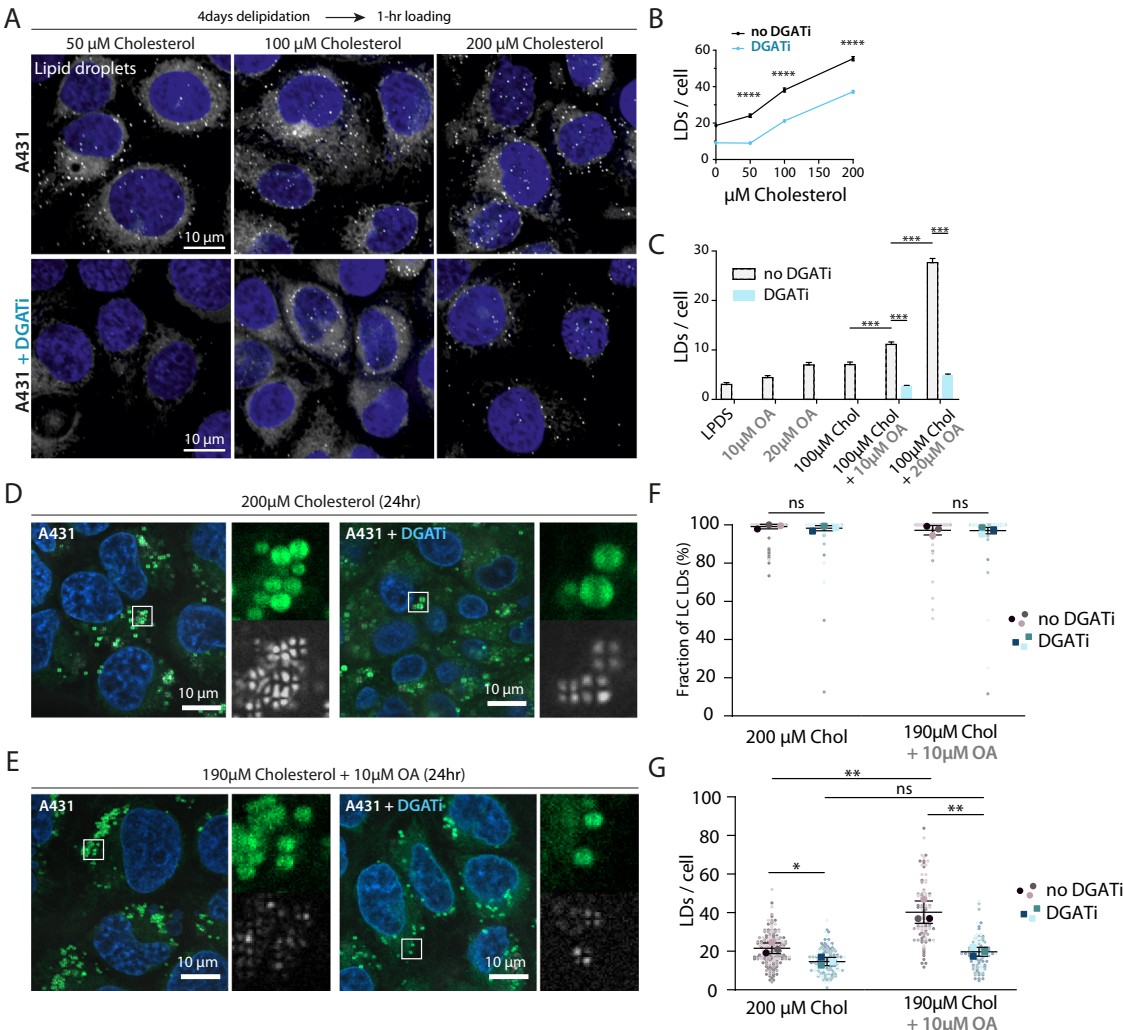

**Fig. 2 | TGs stimulate cholesterol lipid droplet assembly in cells. A** Maximum intensity projections of confocal images of lipid droplets. A431WT cells starved in 5% LPDS for 4 days + DGATi for overnight and loaded with indicated concentrations of cholesterol/cyclodextrin for 1 hour ± DGATi. Cells were fixed and stained with LD540 and DAPI. Scale bar = 10 μm. The experiment was independently repeated three times with similar results. **B** Quantification of lipid droplet numbers per cell. $n$ = 899 cells for 0 μM, 944 cells for 0 μM + DGATi, 734 cells for 50 μM, 1016 cells for 50 μM + DGATi, 547 cells for 100 μM, 948 cells for 100 μM + DGATi, 705 cells for 200 μM, and 864 cells for 200 μM + DGATi condition. Mean ± SEM. ****$p$ < 0.0001, two-tailed student's $t$ test. **C** A431WT cells loaded with cholesterol, oleic acid, or both cholesterol and oleic acid with indicated concentrations for 30 min ± DGATi after similar starvation as in **A**. Quantification of lipid droplet numbers per cell from confocal images of fixed cells stained with LD540 and DAPI. $n$ = 631 cells for LPDS, 899 cells for 10 μM OA, 1175 cells for 20 μM OA, 951 cells for 100 μM chol, 1471 cells for 100 μM chol + 10 μM OA, 929 cells for 100 μM chol + 20 μM OA, 1051 cells for 100 μM chol + 10 μM OA + DGATi, and 870 cells for 100 μM chol + 20 μM OA + DGATi condition. Mean ± SEM. In 20 μM OA + 100 μM chol + DGATi, more CEs are synthesized compared to 10 μM OA + 100 μM chol + DGATi (see Supplementary Figure 2 F), in line with more LDs

being formed. ****$p$ < 0.001, two-tailed student's $t$ test. **D** A431 cells imaged after 24 h of cholesterol (200 μM) or cholesterol (200 μM) and DGAT inhibitors feeding. Bodipy was added upon imaging for LD labeling and analysis (**F**). **E** A431 cells imaged after 24 h of cholesterol (190 μM) + oleic Acid (10 μM) or cholesterol (190 μM) + oleic acid (10 μM) and DGAT inhibitors feeding. Bodipy was added upon imaging for LD labeling and analysis (**G**). **F** Analysis of **D** and **E**. Fraction of liquid crystalline LDs, Mean ± SD. $N$ = 65, 54, and 39 cells for 200 μM cholesterol. $N$ = 49, 42, and 37 cells for 200 μM cholesterol + DGATi. $N$ = 40, 20, and 34 cells for the 190 μM cholesterol + 10 μM OA. $N$ = 41, 27, and 26 for 190 μM cholesterol + 10 μM OA + DGATi. Each color point represents a data point from a replicate. The experiment was independently repeated three times with similar results. ns $p$ = 0.4647 & $p$ = 0.9444, no significant differences two-tailed Nested $t$ test. **G** Analysis of **D** and **E**. Number of LDs per cell, Mean ± SD. $N$ = 65, 54, and 39 cells for 200 μM cholesterol. $N$ = 49, 42, and 37 cells for 200 μM cholesterol + DGATi. $N$ = 40, 20, and 34 cells for the 190 μM cholesterol + 10 μM OA. $N$ = 41, 27, and 26 for 190 μM cholesterol + 10 μM OA + DGATi. Each color point represents a data point from a replicate. The experiment was independently repeated three times with similar results. (Left to Right) *$p$ = 0.027, **$p$ = 0.0065, ns $p$ = 0.0547, **$p$ = 0.0047 two-tailed Nested $t$ tests.

argues that CE LD nucleation was enhanced by ongoing TG synthesis, despite the fact that no OA was added to the medium. At higher concentrations, i.e., 100 and 200 μM cholesterol, LDs were made in both WT and DGAT-inhibited conditions, but the number and integrated size of LDs were significantly lower in the latter case (Fig. 2B, Supplementary Figure 2C). These data support the hypothesis that CE LD nucleation is facilitated by ongoing TG synthesis, even though no OA was supplied to the cells.

Based on the above observations, we investigated if spiking cells with small amounts of OA during cholesterol loading might ease CE LD formation by TG synthesis. While 100 μM cholesterol alone for 30 min did not efficiently induce LD formation (Fig. 2C, Supplementary Figure 2D, E), adding 10 μM OA together with 100 μM cholesterol increased CE LD amounts more than what was achievable by 100 μM cholesterol or 10 μM OA alone (Fig. 2C, Supplementary Figure 2D). The effect was even more pronounced with 20 μM OA, which when spiked

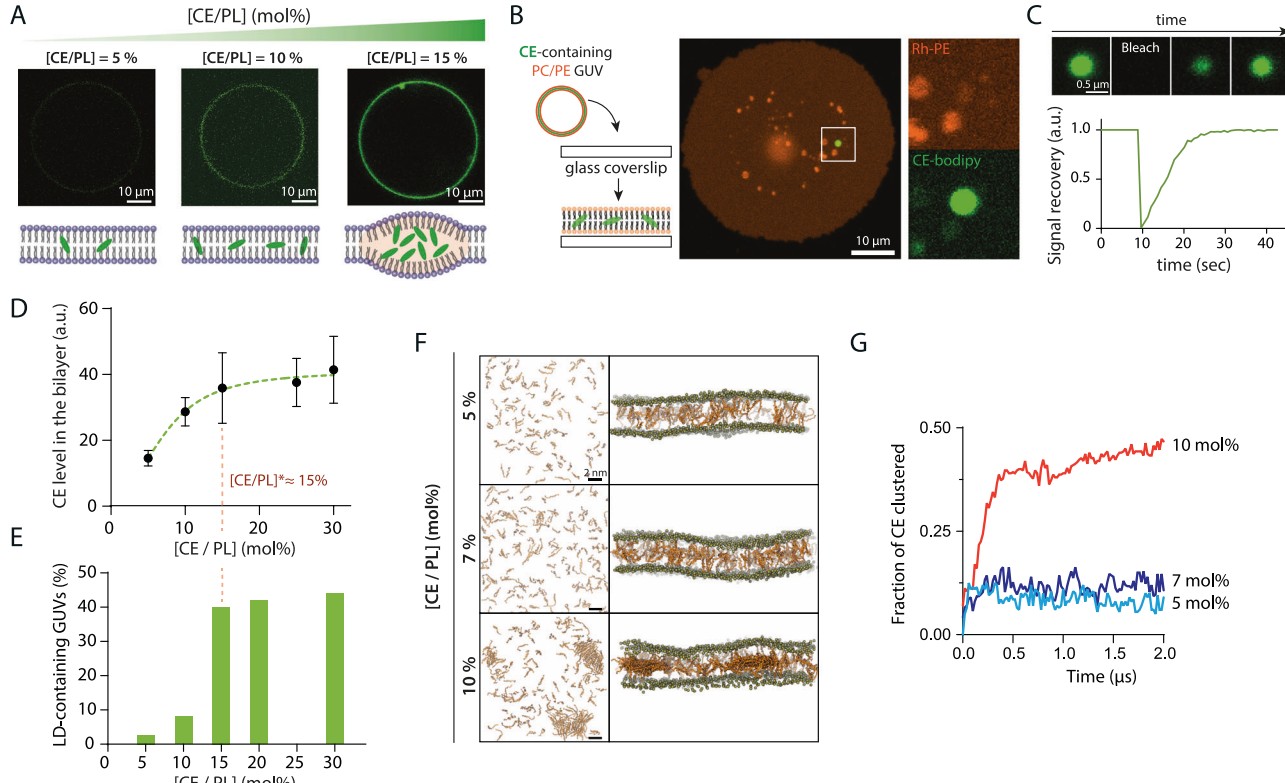

**Fig. 3 | CEs condense into droplets in model membranes. A** GUVs were made with different concentrations of CE (here, cholesteryl oleate) and imaged under confocal microscopy. Top: images of gradually CE-concentrated GUVs. CE condensates started appearing from the concentration of 15 mol%. Bottom: Schematic representation of gradually increasing CE concentration within GUV bilayer. **B** Left: Schematic representation of GUV splashing on a bare glass coverslip. Right: microscope image of a splashed GUV, appearing flat on the glass surface. CE condensates were still visible and immobilized. **C** Signal recovery in a totally bleached CE condensate from splashed GUV as shown in **B**. The signal was normalized by the initial fluorescence signal. **D** Quantification of the amount of free CE in the membrane by measuring the CE/PL fluorescence ratio in splashed GUVs as shown in **C**. Points: Mean ± SD, 5–10 GUVs per experimental point. **E** Quantification

of the occurrences of droplet-containing GUVs according to the [CE/PL] ratio. Above the [CE/PL] ratio of 15%, both the CE enrichment in the membrane and the occurrences of droplet-containing GUVs appear to be almost constant, suggesting that the critical concentration is around 15%. $N = 9–37$ GUVs per concentration. **F** Representation of CE distribution in a model ER bilayer at the end ($t = 2\,\mu s$) of the simulation period. Top-view (left) and side-view (right) representation of the system. CE molecules are shown in orange. Phosphate atoms of the membrane lipids are shown in tan in the side-view representation. Other components of the system (membrane lipids, water, and ions) are not shown for clarity. **G** Plot showing the fraction of CE molecules clustered in the bilayers during the simulation. The remaining CE molecules that are not bound to clusters are individual monomers. The data have been averaged over three simulation repeats.

together with 100 μM cholesterol, increased the number of CE LDs more than 2-fold (Fig. 2C, Supplementary Figure 2D–F). Importantly, the effect of OA spiking was sensitive to DGAT inhibitors indicating that the increased number of LDs was achieved via enhanced TG synthesis. These experiments strongly support the idea that ongoing TG synthesis mediates the efficient nucleation of CE LDs.

To further validate these findings, we used a long cholesterol loading time to enable the formation of large LDs that could be visualized under polarized light. A431 cells were fed with 200 μM cholesterol for 24 h, with or without DGAT inhibitors (Fig. 2D). The number of LDs under the crystalline phase was unchanged (Fig. 2D, F), i.e., LDs were >90% CE rich at 37 °C (Fig. 1F, Supplementary Figure 2G). However, the number of such LDs was decreased by almost half in the DGAT-inhibited condition (Fig. 2G), although no OA was added. This result is consistent with the decrease in the number of nucleated LDs in the presence of DGAT inhibitors (Fig. 2A–C). Next, we modified the experiment by loading with 190 μM cholesterol spiked with 10 μM OA, representing 5% of the load (Fig. 2E). The LDs generated still exhibited the crystalline organization (Fig. 2F, Supplementary Figure 2I), indicating that the level of OA supplied and the TGs subsequently made did not alter the physical state of the LDs. Consistently with TG spiking in the nucleation experiment (Fig. 2C, Supplementary Figure 2D), 5% OA supply was sufficient to significantly increase the number of CE LDs (Fig. 2G, Supplementary Figure 2H). Moreover, the presence of DGATi

did not change the physical state of the LDs but decreased the number of LDs to a similar level as without OA (Fig. 2G).

Our data indicate that ongoing TG synthesis determines the efficiency of CE LD formation. To rule out that these findings were specific to the A431 cell line, we also analyzed Cos7 and HeLa cells, loaded with cholesterol or cholesterol and OA. There, we likewise found that the inhibition of DGATs significantly reduced the number of LDs (Supplementary Figure 2J–R).

## CE LDs nucleate at 10–15% to membrane phospholipids

Based on the above data, we hypothesized that CE molecules might not efficiently condense into droplets in membranes and that TGs could assist in this step. We next investigated the potential principles underlying this phenomenon.

To investigate the behavior of CEs in a simple bilayer, we mixed dioleoylphosphatidylcholine (DOPC) and dioleoylphosphatidylethanolamine (DOPE) phospholipids (70/30), reported by rhodamine-PE at 1%, and cholesteryl oleate (CE), reported by cholesteryl linoleate-NBD at 1%. The mixture was then dried and hydrated to make giant unilamellar vesicles (GUVs) with different CE/phospholipid ratios by electroformation. Up to 10% CE, we observed a uniform CE signal in membranes (Fig. 3A, Supplementary Figure 3A). Only at a concentration of 15–20%, CE droplets were seen in the GUV bilayer (Fig. 3A, Supplementary Figure 3A). To validate this

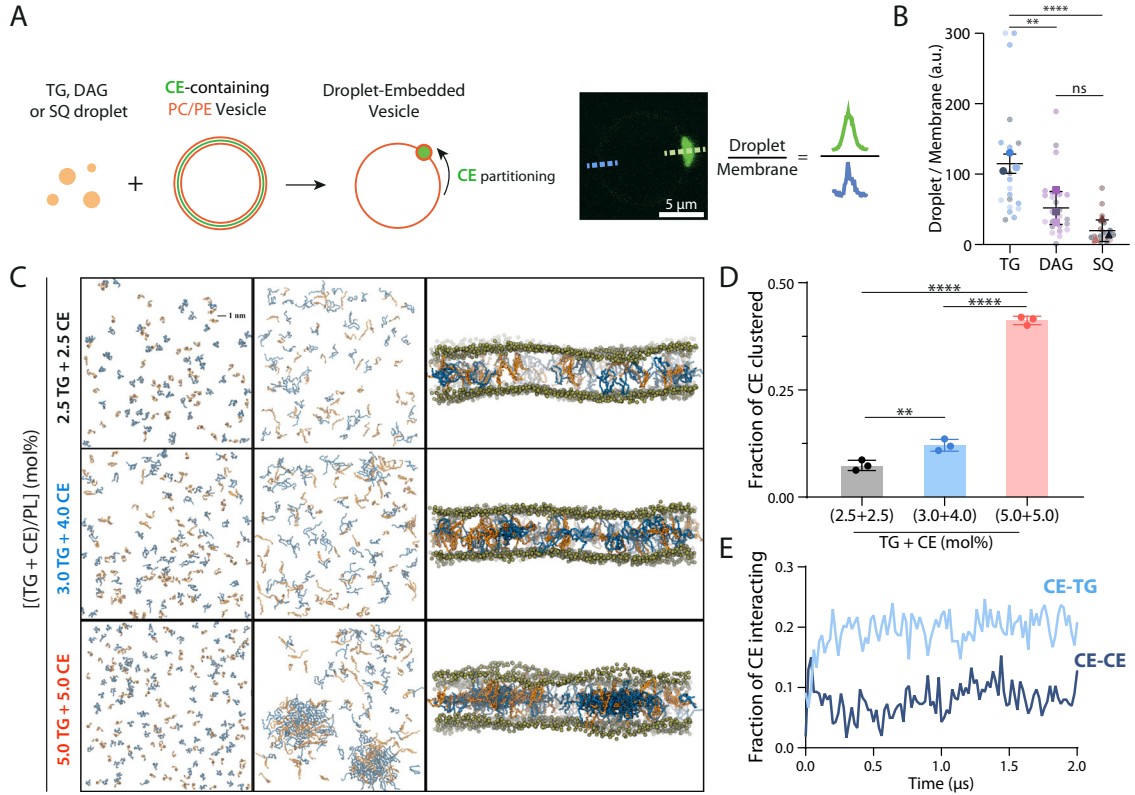

**Fig. 4 | TGs facilitate CE clustering and incorporation into droplets. A** Left: Schematic representation of droplet-embedded vesicles (DEVs) protocol. An emulsion of either triolein (TG), dioleoylglycerol (DAG), or squalene (SQ) was mixed with CE (cholesteryl oleate)-containing (10%) PC/PE (70/30) GUVs. The droplets go in between the two layers of the GUVs, forming a DEV. The CE already present in the membrane can then diffuse to the embedded droplet. Right: representation of the droplet/membrane partitioning ratio of CE: the CE-bodipy fluorescence in the embedded droplet was compared to the CE-bodipy fluorescence in the membrane. **B** Quantification of **A** for TG, DAG, and SQ DEVs. Mean ± SD. 5, 9, and 5 DEVs for TG, 6, 10, and 8 for DAG, and 5, 7, and 8 for SQ conditions. Each color point represents a data point from a replicate. The experiment was independently repeated three times with similar results. (Left to Right) **p = 0.0044, ****p < 0.0001, ns p = 0.1145, two-tailed Nested t tests.

**C** Representation of CE (orange) and TG (blue) distribution in a model ER bilayer at 2.5 mol% of each neutral lipid; at TG-CE concentration of 3 mol%–4 mol% respectively and at 5 mol% of each neutral lipid at the beginning ($t = 0$) and at the end ($t = 2\,\mu s$) of the simulation period. Top-view (left and center) and side-view (right) representation of the system. Phosphate atoms of the membrane lipids are shown in tan in the side-view representation. Other components of the system (membrane lipids, water, and ions) are not shown for clarity. **D** Plot showing the fraction of CE molecules clustered in the bilayers during the simulations. The data have been averaged over three simulation repeats. Mean ± SD. **E** Plot showing the fraction of CE molecules interacting with CE molecules (CE-CE) and CE molecules interacting with TG molecules (CE-TG) in 3% TG + 4% CE system. The data have been averaged over three simulation repeats.

observation, we burst the GUVs on a glass coverslip to better visualize the membrane and the droplet, in 2D. Only starting at 15%, we detected the presence of CE droplets in the membrane (Fig. 3B). The droplet signal recovered after photobleaching, indicating that the droplet was in equilibrium with CE molecules in the bilayer (Fig. 3C). Accordingly, above 15%, the concentration of free CE in the bilayer was almost constant, the excess being likely adsorbed by the nucleated droplet (Fig. 3D); in parallel, the frequency of GUVs containing droplets sharply increased at 15% (Fig. 3E). These data indicate that CE droplets nucleate in DOPC/DOPE (70/30) bilayers at 15-20% concentration of CE/phospholipids.

To further investigate the molecular details of CE nucleation, we examined the process using atomistic umbrella sampling simulations. We performed simulations with varying concentrations of CE (5 mol%, 7 mol%, and 10 mol%) in phospholipid bilayers having a lipid composition similar to that of the ER (see Methods and ref. [14]). In line with the GUV experiments, we failed to observe a stable association between the CE molecules at 5-7 mol% concentration (Fig. 3F, G). However, when the CE concentration in the bilayer was increased to 10 mol%, we observed CE molecules to form stable aggregates (Fig. 3F, G). Similar CE clustering was observed at higher CE concentrations in DOPC/DOPE (70/30) bilayers (Supplementary Figure 3D), which matches the lipid composition used in the experiments. The arrangement of CE

molecules in these aggregates was such that primarily the sterol rings were stacked against each other (Supplementary Figure 3B, C), which is reminiscent of the liquid crystal organization.

## TGs facilitate CE clustering and LD formation in the bilayer

The nucleation concentration for CE, 10–15%, is much higher than the one observed for TG, 3–4%, for a similar bilayer composition[15–17]. Thus, CE condenses with more difficulty into droplets than TG. To investigate if CEs can be favorably incorporated into a nascent TG droplet in a bilayer, we generated GUVs containing 10 mol% of CE molecules, i.e., when no CE droplet formed (Fig. 3D, E). Then, we incorporated artificial droplets to make droplet-embedded vesicles[37–39] with different neutral lipid compositions and determined the partitioning of CEs between the bilayer and the droplet (Fig. 4A). We found that CEs partitioned more favorably into TG droplets than diacylglycerol or squalene droplets (Fig. 4B, Supplementary Figure 4A). This analysis suggests that CE LD assembly could be particularly helped by TG pre-clusters and not DAG clusters. In agreement with this conclusion, the inhibition of DGATs which accumulates DAG, was inhibitory to LD nucleation (Fig. 2).

To directly test our hypothesis, we used all-atom molecular dynamics simulations to capture nucleation in the case of TG/CE mixtures. We explored the mixing process of CEs with TGs in a realistic

ER bilayer containing both TGs and CEs at varying concentrations. At low equimolar concentrations of both neutral lipid species (2.5 mol% CE + 2.5 mol% TG), we did not observe the formation of a stable cluster by either of them (Fig. 4C). However, when the concentration of TG was raised to 3 mol% and CE to 4 mol%, TG molecules began to form a stable cluster with CEs (Fig. 4C), with CE and TG molecules interspersed in the cluster. At a higher concentration (5.0 mol% CE + 5.0 mol% TG), a larger fraction of CE molecules clustered with TGs (Fig. 4C). Analysis of the relative populations of different species showed that the fraction of CEs clustering with TGs increased during the mixing process (Fig. 4D), while the fraction of free CE monomers decreased (Supplementary Figure 4B, C). For comparison, very little change was observed in the fraction of CEs embedded in TG-free clusters of CEs (Supplementary Figure 4D), suggesting that TGs facilitate the incorporation of CEs into neutral lipid droplets. Interestingly, while at 7% CE nucleation did not happen (Fig. 3F, G), a mixture of 3% TG and 4%CE (a total of 7% of neutral lipids), allowed the nucleation of CE-containing droplets (Fig. 4E). Altogether, these data argue that it is more difficult to incorporate a CE molecule in a pure CE pre-cluster than in a TG pre-cluster, which can explain the difficulty of nucleation in the case of CE. Hence, TG pre-clusters catalyze the nucleation of CE droplets better than CE pre-clusters.

## TG synthesis is required for the efficient nucleation of CE LDs by seipin

Since seipin clusters TGs, thereby presumably controlling the nucleation of TG-rich LDs, and TG clusters stimulate CE LD nucleation, we speculated that CE LDs should preferentially form at seipins as well. To address this, we employed A431 cells with endogenously tagged seipin-GFP and loaded them with cholesterol from MßCD. We found that seipin colocalized with the formed CE LDs (Fig. 5A). To assess if seipin affects the sites of CE LD formation, we employed A431 cells where endogenous seipin was trapped in the nuclear envelope[12]. In these cells, CE LDs formed more readily at this ER subdomain (Fig. 5B, Supplementary Figure 5A). Together, these results indicate that seipin associates with CE LDs and can control the sites of CE LD formation in A431 cells.

In silico experiments indicate that, mechanistically, seipin controls TG LD nucleation by decreasing the nucleation concentration of TGs from 3–4% to 1.25% of bilayer lipids, via the interaction of TGs with seipin transmembrane and luminal helices[9,10,13–16]. We, therefore, investigated if seipin also interacts with CEs using atomistic simulations. In a model ER bilayer with the seipin oligomer (luminal domain + transmembrane region) and 5 mol% CE randomly distributed around it, we observed diffusion of CEs into the lumen of the seipin oligomer and subsequent interaction of the carbonyl group with residue S166 or S165 in the membrane-embedded α2-α3 helices (Fig. 5C, D, Supplementary Figure 5B), previously reported as TG interaction sites[13,14] and recently found also to interact with CE[40]. Mutations of these residues to A abrogated the interaction (Fig. 5E), as shown for TGs[13,14]. We further observed incoming CE molecules to interact with a seipin-bound CE and to form dimers (Supplementary Figure 5B), reminiscent of the CE arrangement in the liquid crystalline phase.

We then asked if the interaction strengths of seipin with TG and CE differ, using a long timescale (5 μs) simulation of only the seipin oligomer luminal domain embedded in a model ER bilayer with 2.5 mol% TGs and 2.5 mol% CEs. This showed that TG and CE interacted with S166 residues on different protomers (Supplementary Figure 5C). However, we did not observe instances of one neutral lipid species displacing the other from its S166-bound state. We also performed umbrella sampling atomistic molecular dynamics simulations to calculate the free energies of TG-S166 and CE-S166 interactions. This revealed that, within the error bars, TG and CE manifest similar free energy profiles (Supplementary Figure 5D): the bound state (about 0.3 nm) is separated from the bulk state by a barrier of about 8 kJ/mol.

Therefore, the binding affinity of TG and CE with these key residues is essentially similar.

The above results indicate that seipin can similarly interact with TG and CE and, therefore, one might expect TG and CE formation in cells to occur rather similarly at seipin-defined sites. However, although TG and CE could interact similarly with seipin, a condensation step must occur for the molecules to subsequently form a nascent droplet. Such condensation happens efficiently for TG, at ~2–3 mol%[14–17], while for CE it happened only at 10–15 mol% (Fig. 3) in model membranes. Therefore, in the presence of seipin, the clustering of TGs at seipins could facilitate the subsequent condensation and nucleation of CE LDs.

To disentangle the impact of TGs and seipin in the condensation of CEs into LDs, we investigated cells under conditions where either TG synthesis was blocked and/or seipin removed. We have earlier reported that when A431 cells were loaded with OA, first TG LDs appeared within a few minutes[12] and in 20 min, tens of droplets were detectable OA concentration-dependently, with up to ~50 LDs at 200 μM OA[14]. However, when we loaded A431 cells with cholesterol from MßCD with similar lipid concentrations, CE LDs appeared considerably more slowly, with ~50 droplets detectable only in 1 h using 200 μM cholesterol (Figs. 2A, 5G). Interestingly, when seipin knockout cells were similarly loaded with cholesterol, the initiation of new LDs required a higher cholesterol concentration (100 μM vs. 50 μM in WT cells) but at 200 μM cholesterol, the LD numbers were essentially similar to WT cells. Moreover, the numerous tiny LDs characteristic of seipin knockout cells upon OA loading (~250 LDs with 200 μM OA in 1 h[14],) were not observed upon cholesterol loading (Fig. 5F, G). These data argue that despite the ability of seipin to interact with CEs and position CE LDs in cells, CEs behave fundamentally differently from TGs in the ER, and that this contributes to CE LD formation in cells.

Finally, we analyzed the effect of inhibiting DGATs during CE LD formation in seipin-deficient cells. Remarkably, the effect of DGAT inhibitors was similar in WT and seipin knockout cells (Fig. 5F, G). This indicates that the deficiency of seipin brought no additional impact to CE LD formation beyond what was achieved by inhibition of TG synthesis alone. Taken together, these results suggest that seipin affects CE LD formation via TGs: TGs cluster more readily than CEs in the bilayer and associate with seipin, which in turn helps to co-cluster CEs and promote CE LD formation.

## Discussion

CEs form metastable supercooled liquids and at high CE concentrations, supercooled smectic liquid crystalline droplets. They tend to crystallize, even at body temperature. Such crystallization could be relevant to cholesterol-related disorders such as atherosclerosis, fatty liver disease, and Niemann-Pick type C disease[41–43]. Yet, it seems that cells manage to efficiently store CEs in metastable liquid forms in LDs, and prevent crystal formation.

At 37 °C, the liquid crystalline phase appears in droplets when the CE-to-TG ratio exceeds 9/1, a threshold that decreases with temperature (Fig. 1F)[32]. In this regime, CE molecules are highly ordered and form radial lamellar phases[32–34], likely arising from CE-CE dimerization capacity (Supplementary Figure 3B). In the ER bilayer, these characteristics of CE molecules would reduce the freedom of CEs and possibly retard their arrangement to condense in a droplet. This would explain the delayed nucleation of CE droplets in model membranes, at 10-15% (CE/phospholipids) compared with 3–4% for TG. This implies that if CEs are not removed from the ER membrane they would reach a high concentration, which could be deleterious to the membrane.

Given the particularity of the crystalline phase, CEs should require specific membrane settings for condensing into LDs. We found that TG, and not diacylglycerol, for example, fulfills this need by acting as a catalyst, precipitating the incorporation of CEs into nascent LDs. Indeed, with its lower nucleation concentration, TGs can form clusters

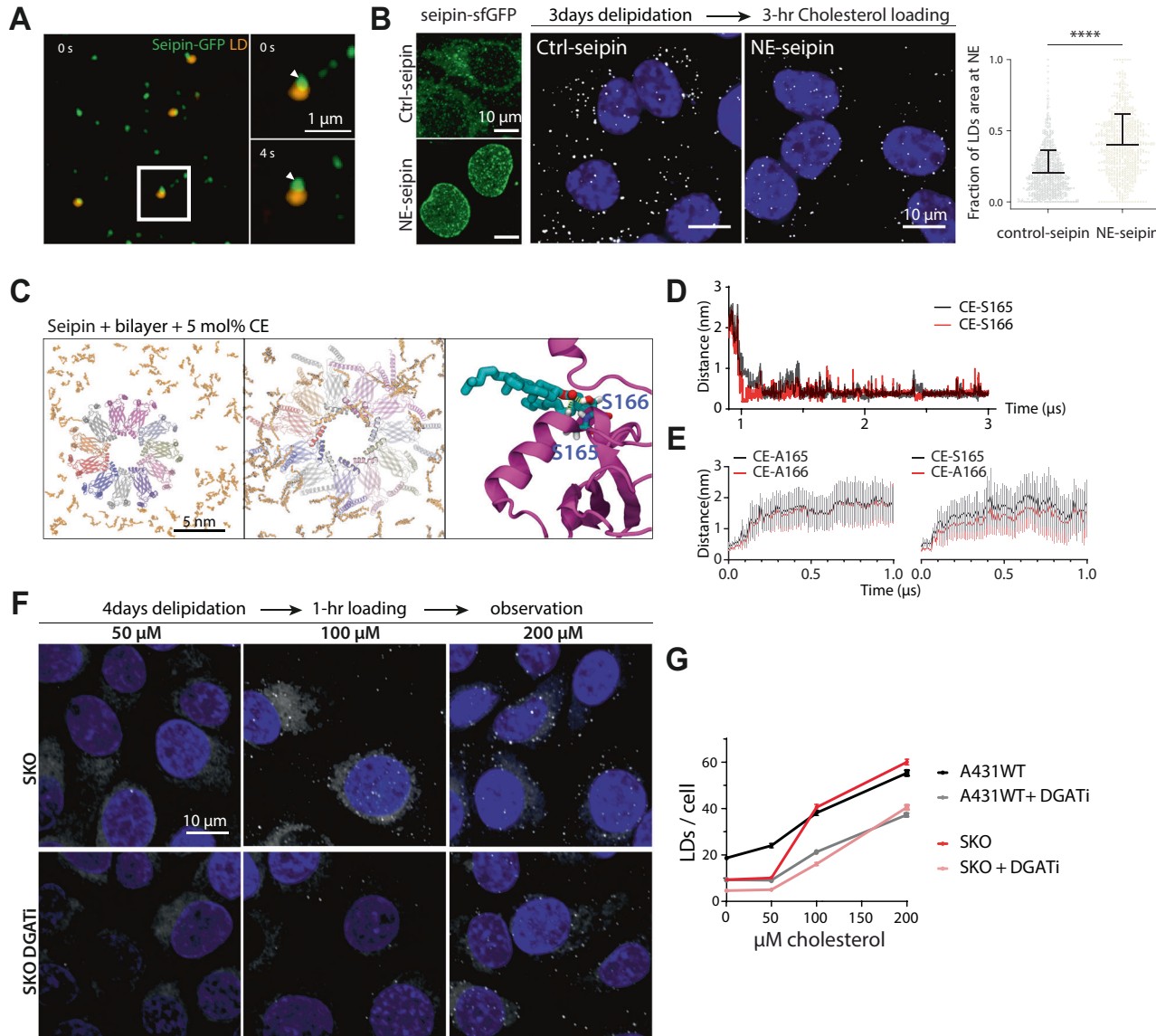

**Fig. 5 | Seipin controls CE LD nucleation sites via its TG clustering capacity.**
**A** Airyscan images of cells with endogenously GFPx7-tagged seipin starved for
3 days in 5% LPDS + DGATi for overnight, washed and loaded with 100 μM cho-
lesterol cyclodextrin for 90 min, stained with Autodot to visualize lipid droplets,
and imaged live. Arrowheads indicate a stable contact between seipin and a lipid
droplet during 4 seconds. Scale bar = 1 μm. The experiment was independently
repeated three times with similar results. **B** Maximum intensity projections of
deconvolved widefield images of seipin-sfGFP in control and seipin NE-trap cells.
(Left) Confocal images of lipid droplets in control and seipin NE-trap cells. (Right)
Cells were starved for 3 days in 5% LPDS + DGATi overnight and loaded with 200 μM
cholesterol cyclodextrin + DGATi for 3 h, fixed and stained with DAPI and LD540.
Scale bar = 10 μm. LD area overlapping with nuclei quantified as fraction of LDs area
at NE. Control $n = 727$ and NE-trap $n = 671$ cells. Mean+ SD and all individual data
points. ****$p < 0.0001$, two-tailed student's $t$ test. An independent similar experi-
ment presented in Supplementary Figure 5 A. **C** Top-view representation of CE
molecules (5 mol%, shown in orange) around the seipin oligomer associated with a
model ER membrane. Left: At the beginning ($t = 0$). Middle: At the end ($t = 3$ μs) of
the simulation period (zoomed in to show CE association with α2−α3 helices). Each
protomer is shown with a different color. Right: A close-up view showing the

interaction of a CE molecule with the residues S165 and S166 on a seipin protomer.
Yellow bars represent the interacting atoms between the residues and CE.
**D** Minimum distance between a tagged CE molecule and the residues S165 and
S166 of a single protomer. **E** Left: Following the S165A-S166A mutation, the mini-
mum distance between the tagged CE molecule (previously bound to S166) and the
residues A165 and A166 of the protomer in which the CE was bound. The data have
been averaged over three simulation repeats. Right: following the S166A mutation,
the minimum distance between the tagged CE molecule (previously bound to
S166) and the residues S165 and A166 of the protomer in which the tagged CE was
bound. The data have been averaged over three simulation repeats. **F** Maximum
intensity projections of confocal images of Seipin-KO (SKO) cells starved in 5%
LPDS for 4 days + DGATi for overnight and loaded with indicated concentrations of
cholesterol cyclodextrin for 1 hour ± DGATi. Cells were fixed and stained with
LD540 and DAPI. Scale bar = 10 μm. **G** Quantification of lipid droplet number
per cell. $n = 905$ cells for SKO LPDS, 1013 cells for SKO LPDS + DGATi, 946 cells for
SKO 50 μM, 1084 cells for SKO 50 μM chol + DGATi, 918 cells for SKO 100 μM, 1009
cells for SKO 100 μM + DGATi, 1048 cells for SKO 200 μM, and 895 cells for SKO
200 μM + DGATi condition. Mean ± SEM, including data from (2B).

that more favorably incorporate CE molecules than CE clusters do. We
speculate that the critical TG concentration needed to catalyze CE LD
nucleation is the one preventing the formation of a crystalline phase in
the ER, i.e. 1/9 of TG/CE at 37 °C. Afterward, a nucleated CE-rich LD

grows by acquiring more CE molecules and thereupon enters the
liquid crystalline regime.
    As the prime TG-clustering factor[8,10,13,14], seipin can control CE LD
formation sites, as long as TG molecules are available in the ER. Indeed,

by interacting with TGs, seipin forms TG pre-clusters that can recruit CEs to nucleate CE LDs. Accordingly, the inhibition of DGATs severely decreased the number of nucleation events, as fewer TGs would be available. In agreement with this model, in *Saccharomyces cerevisiae*, CE LDs form at seipins but, when TG accumulation is completely inhibited by the removal of Lro1 and Dga1, the sole enzymes making TGs, few LDs form away from seipins[28]. Similarly, retinyl palmitate or squalene is better incorporated into LDs when TG synthesis is ongoing[26,28].

Since the accumulation of neutral lipids would be deleterious to the ER membrane[27] and hence, overall cell functioning, removing them rapidly via LD formation is critical. Based on cell observations, in vitro, and in silico data, TG seems to be so far the neutral lipid with the lowest nucleation concentration; it is rapidly storable in LDs, possibly reflecting why it is the most ubiquitous neutral lipid across many living systems. Therefore, accompanying TG synthesis with the synthesis of other neutral lipids of lower nucleation efficiency might represent a strategy for cells to ensure efficient nucleation of the latter. At seipins where TGs are pre-clustered, these neutral lipids would better interact with TGs and nucleated LDs: the initial TG seeds will act as thermodynamic pumps, retrieving other neutral lipids from the bilayer. In this view, the role of seipin is coupled to TG, as seen in yeast[28]. This proffers an underappreciated role to TGs in LD formation, beyond their traditional energy storage function.

Based on our data, mammalian cells must keep a background of TG production in the ER membrane during lipogenic periods, even if TGs were not the primary neutral lipid stored. This TG seed would play a critical role in nucleating LDs enriched in the synthesized neutral lipid, at seipins. Such a surveillance role was well illustrated when the cells were supplied with a tiny amount of oleic acid, to generate TGs under conditions of high cholesterol feeding. There, many more CE-rich LDs were made as compared with no oleic acid-fed cells. The proposed function of TG on CE LD assembly likely depends on the cell physiology and may be critical for cell types that may have more difficulty in handling CE storage. For instance, Cos7 cells strongly relied on TG synthesis to solubilize CEs, as they were more sensitive to DGAT inhibition and barely assembled CE LDs under this condition (Supplementary Figure 2J–O). In agreement with this, Cos7 cells did not display any liquid crystalline organization in CE LDs upon cholesterol feeding, as opposed to HeLa cells (Supplementary Figure 2P–R).

Finally, cells may systematically coordinate the level of TG synthesis when other neutral lipids are made. Such a strategy would enable them to rapidly remove the latter from the ER bilayer via the capacity of TG, better than other neutral lipids, to efficiently cluster and form droplets. In this view, seipin would globally define the formation sites of LDs via clustering TG.

## Methods
### Materials
DAPI (Sigma, D9542), LD540 (Princeton BioMolecular Research), Autodot (Abgent, SM1000b), DGAT 1 inhibitor (Sigma PZ0207), DGAT2 inhibitor (SigmaPZ0233). Cell culture reagents and general reagents were purchased from Gibco/Thermo Fisher, Lonza, and Sigma-Aldrich. Lipoprotein-deficient serum (LPDS) was made from fetal bovine serum (FBS) as previously described[44]. Methyl-ß-cyclodextrin for cell culture (#C4555) was purchased from SigmaAldrich, cholesterol (Avanti, 70000 P) was purchased from Avanti.

The cholesterol/methyl-ß-cyclodextrin solution was prepared at a final concentration of 1 mM of cholesterol as following. A suitable amount of methyl-ß-cyclodextrin was dissolved in cell culture media and then incubated with crystal cholesterol at 1/20 molar ratio (cholesterol/ methyl-ß-cyclodextrin) for 24 h with agitation at 37 °C. The resulting solution was filtered using 0.2 μm syringe filter and conserved at 4 °C until use.

For 1 mM of oleic acid-containing media preparation, 10% of BSA solution in DPBS (#A1595 SigmaAldrich) was mixed with cell culture media at 1/10 v/v ratio. The resulting solution was incubated with pure oleic acid (#O1383 SigmaAldrich) to obtain a final concentration of 1 mM. The resulting mixture was vortexed and then sonicated for a few minutes and incubated at 37 °C for one hour. The resulting solution was filtered using 0.2 μm syringe filter and conserved at 4 °C until use.

### Cell culture and lipid manipulations
A431 cells (ATCC, Cat# CRL-1555, RRID:CVCL_0037) were maintained in Dulbecco's Modified Eagle's Medium (DMEM) supplemented with 10% FBS, penicillin/streptomycin (100 U/mL each), and L-glutamine (2 mM) at 37 °C in 5% CO$_2$. All cell lines were regularly tested negative for mycoplasma infection by PCR. For lipid droplet imaging experiments cells were delipidated by 3- to 4-day treatment with serum-free medium supplemented with 5% LPDS. For a more stringent delipidation cells were treated with DGAT1&2 inhibitors (5 μM each) for the final 18 h of delipidation where indicated. DGATi indicates treatment with both DGAT 1 and DGAT2 inhibitors. Following delipidation, cells were loaded with methyl-ß-cyclodextrin complexed cholesterol in 5% LPDS for indicated concentrations and times or with oleic acid in complex with BSA in 8:1 molar ratio prepared as described[45] in 5% LPDS.

For 24 h lipid loading experiments, COS7, HeLa, and A431 cells were maintained in High Glucose with stabilized Glutamine and with Sodium Pyruvate Dulbecco's modified Eagle's Medium (DMEM) (Dutscher) supplemented with 10% fetal bovine serum and 1% penicillin/streptomycin at 37 °C and 5% CO$_2$. Cells were plated in Mattek dishes (#P35G-0-20-C) for 24 h, then incubated with the culture media containing cholesterol, oleic acid, or both of them at the indicated concentration and time. When mentioned, DGAT 1 and DGAT2 inhibitors were added at 5 μM during lipid treatment.

### Stable cell lines
A431 Seipin-KO (SKO) cells were generated with CRISPR/Cas9 technology as described[23]. The SKO clone in this study corresponds to S2AB-15 in[23]. Generation of endogenously tagged seipin-GFPx7 cells and generation of seipin-sfGFP control and seipin NE-trap cells are previously described[12]. To generate end-seipin-GFPx7 cells, endogenous seipin was tagged with GFP11x7 (Addgene #70224, a gift from Bo Huang), which becomes fluorescent upon self-complementation with a non-fluorescent GFP1-10 fragment expressed in the same cell[46]. We utilized a co-selection strategy for simultaneous tagging of endogenous seipin and integration of the cassette overexpressing GFP1-10 into the AAVS1/Safe Harbor locus through homology-directed repair. Three plasmids (1: Seipin-GFP11x7 homology-directed repair template; 2: GFP1-10 overexpression cassette with puromycin selection marker on AAVS1 integration template; 3: Cas9, sgBSCL2/seipin, sgAAVS1 overexpression plasmid) were transfected at 5:1:4 ratio into A431 cells. Selection was done with puromycin and single clones were isolated using limiting dilution and based on GFP fluorescence. GFP1-10 fragment was codon-optimized and synthesized by Genescript because the plasmid (Addgene #70219[46], a gift from Bo Huang) was of low codon adaption index (CAI, analyzed at www.genscript.com/tools/rare-codon-analysis) and was poorly expressed by us in human cells. Homozygous knock-in validated by genomic PCR and by western blot with an in-house-generated antibody against seipin and an antibody against GFP.

### Cell stainings
For confocal and widefield imaging of fixed cells, cells were washed with PBS and fixed with 4% PFA in 250 mM Hepes, pH 7.4, 100 μM CaCl$_2$, and 100 μM MgCl$_2$ for 20 minutes. Subsequently, cells were washed with PBS, quenched in 50 mM NH$_4$Cl for 15 minutes, and washed with PBS. To stain lipid droplets and nuclei, cells were

incubated for 30 minutes with LD540 (1 µg/mL) and DAPI (5 µg/mL) in PBS at room temperature. After staining, cells were washed with PBS and imaged in PBS. For live Airyscan imaging, cells were grown on a LabTek II #1.5 glass-bottom dish-coated with 10 µg/mL fibronectin. Lipid droplets were stained with Autodot (0,1 mM) for 5 min and imaged in Gibco FluoroBrite DMEM supplemented with 5% LPDS at +37 °C, 5% $CO_2$.

## Imaging and image analysis

For lipid droplet analysis cells were delipidated and cultured on 384- or 96-well high-content imaging plates (Corning) and imaged with Perkin-Elmer Opera Phenix automatic spinning-disk confocal microscope using 63x water objective, NA 1.15. Lipid droplets and seipin-sfGFP in control and NE-trap cells were imaged with Nikon Eclipse Ti-E inverted widefield fluorescence microscope using a 40x air objective, NA 0.75, and 1.5x zoom. Z-stacks were acquired to span whole cells on LD540 and DAPI channels. Image stacks were deconvolved with Huygens batch processing application (https://svi.nl/HuygensSoftware, v. 22.10) and image stacks were maximum intensity projected with custom MATLAB scripts (http://www.mathworks.com/products/matlab/, v. 9.2.0.538062; scripts are available upon request). Lipid droplets were detected with Ilastik (v. 1.3.2)[47] by pixel and object classification utilizing machine learning algorithms and final binary images were used for analysis in CellProfiler[48]. Automatic cell segmentation and automatic image analysis were done with CellProfiler with a previously described protocol[12,49,50]. Cell nuclei were detected with Otsu adaptive thresholding method in DAPI images and the cytoplasm was detected as the faint background on the DAPI channel. All high-content imaging experiments were carried out 2–3 times with similar results, with hundreds of cells analyzed per condition and reported as described[12,14,50]. For Airyscan images cells were imaged with Zeiss LSM 880 confocal microscope equipped with Airyscan detector using a 63x Plan-Apochromat oil objective, NA 1.4. Images were Airyscan-processed automatically with Zeiss Zen software (v. 2.3). Fiji (v. 1.51), CorelDraw (v. 24.1.0.360), and GraphPad Prism (v. 7.04) were used for data visualization.

## Simulation systems

Seipin oligomer (transmembrane helices + luminal domain; or seipin luminal domain only) were taken from our previous work[14]. Membrane systems with varying concentrations of CE (cholesteryl oleate) and TG (triolein) were prepared using the protocol described in our previous work[14]. For seipin simulations, the oligomer was inserted into the membrane systems (see Table 1) using the protocol described in our previous work[14]. In all systems, unless specified, the neutral lipids (CE and TG) were randomly distributed around the seipin oligomer. The resulting systems were solvated and neutralized using counter ions. 0.15 M KCl ions were added to mimic the physiological salt concentration in the cytosol.

## Simulation protocols

Molecular dynamics simulations were performed were performed using Gromacs simulation package v. 2019.6[51]. Charmm36/m forcefield with virtual interaction sites[52–55] was used for the proteins, lipids, water, and counter ions. This was done to enable using a larger integration time step of 4 fs. TIP3P water model was used. The systems were energy minimized, pre-equilibrated under constant pressure and temperature for 1 ns. Nose-Hoover thermostat[56] was used to maintain the temperature at 310 K with a coupling constant of 0.5 ps while pressure was maintained at 1 atm using the Parrinello-Rahman barostat[57] with a coupling constant of 2 ps. Verlet cutoff scheme was used to update the neighbor list every 10 steps. Covalently bonded hydrogen bonds were constrained using LINCS algorithm[58]. A value of 1.0 nm was used to cutoff short-range electrostatic and van der Waals interaction while Particle Mesh Ewald method[59] was used to treat long-

range electrostatic interactions. Production simulations of µs time-scales were performed (see Table 1).

## Umbrella sampling simulations

To investigate the energetics and the mechanism of triglyceride (TG) and cholesteryl ester (CE) binding to seipin, we performed two separate sets of free energy computations using the umbrella sampling method. To simplify the systems for the free energy calculations, we used a seipin monomer instead of the seipin protomer. To this end, TG and CE-bound seipin monomers isolated from our previous simulations[14] and current study respectively, were first re-embedded in a sufficiently large 100 mol% POPC membrane (using a hexagonal prism of size 145 nm × 120 nm). These simplified membrane systems were prepared using the CHARMM-GUI[59,60]. An integration time step of 4 fs was used for the simulations by employing the Hydrogen Mass Repartitioning method[60,61]. After a series of short equilibration simulations, the resulting configurations were used to start pulling simulations, in which both TG and CE were pulled away from the protein in two separate 100 ns simulations at a rate of 0.00005 nm/ps using a spring constant of 10,000 kJ/mol/nm². The lateral distance between the center of mass of the $C_a$ atoms of S165 and S166 and the polar head group atoms for each lipid was used as the reaction coordinates for both pulling and subsequent umbrella sampling simulations. The initial configurations for the umbrella sampling windows were chosen from the pulling simulations at a 0.1 nm interval. A total of ~50 windows were simulated spanning a range of 0.3–5.2 nm along the reaction coordinate. Each window was simulated for 100 ns and a uniform force constant of 1000 kJ/mol/nm² was used in each window. The simulation results were unbiased and the free energy profiles were generated using the Weighted Histogram Analysis Method[62].

## Analysis of simulation data

Standard GROMACS tools were used for analysis. VMD[63] was used for visualizing the trajectories and for rendering images. To calculate the fraction of CEs clustered in the bilayer, we calculated the smallest distance between a chosen CE and the rest of the CE molecules. A distance of 0.35 nm between any two CE molecules was chosen as a criterion to consider the interaction between them. A cluster refers to the interaction between any two molecules. Using the same criteria, we also calculated interaction and cluster formation between CE and TG in mixed neutral lipid simulations. The values were calculated over the entire simulation period and averaged over simulation sets for each system.

## Phase transitions and emulsion experiments

In the phase transitions and emulsion experiments, cholesteryl oleate (CE, Sigma-Aldrich) was heated to 37 °C and 50 °C using hot baths. Emulsion experiments were performed by vortexing for 10 s and sonicating for 10 s, 5 µL of previously liquefied CE (or mix between CE and Triolein (TG, Sigma-Aldrich)) in 70 µL of 50 °C hot HKM buffer (50 mM HEPES, 120 mM Kacetate, and 1 mM $MgCl_2$ (in Milli-Q water) at pH 7.4 and 275 ± 15 mOsm). Silicone oil emulsions were made using the same protocol with the HKM buffer replaced by Silicone oil (100 cst, Sigma-Aldrich). Emulsions were then imaged using polarized confocal microscopy.

## Crystallization experiments

The crystallization experiments were conducted by using a spectro-photometer cuvette as a chamber. The droplets of the emulsion would then go up and flatten at the water-air interface. Video were recorded using a u-eye fast imaging camera on a bresser optical microscope.

## Cholesteryl oleate solubility experiments

In vitro solubility experiments were performed by mixing CE and TO, and CE and Diacylglycerol (DAG, SigmaAldrich) in Eppendorfs. Pictures

**Table 1 | Systems explored using atomistic simulation models (\*LMD: luminal domain of seipin oligomer; \*\*TM + LMD: seipin oligomer with transmembrane helices and luminal domain, \*\*\* see ref. [14] Table S1 for ER membrane lipid composition)**

| Seipin | ER membrane system\*\*\* (mol% of neutral lipids) | Simulation time (in μs) | No. of simulations |
|---|---|---|---|
| No | CE (5 mol%) | 2 | 3 |
| No | CE (7 mol%) | 2 | 3 |
| No | CE (10 mol%) | 2 | 3 |
| No | TG + CE (2.5 mol%+2.5 mol%) | 2 | 3 |
| No | TG + CE (3 mol%+4 mol%) | 2 | 3 |
| No | TG + CE (5 mol%+5 mol%) | 2 | 3 |
| Seipin (TM + LMD\*\*) | CE (5 mol%) | 3 | 1 |
| Seipin (LMD\*) | CE (5 mol%) | 5 | 1 |
| Seipin (TM + LMD\*\*) | CE (2.5 mol% within oligomer lumen and 2.5 mol% in the bulk) | 1 | 5 |
| Seipin (LMD\*) | TG + CE (2.5 mol%+2.5 mol%) | 5 | 1 |
| Seipin (S166A mutant) | CE (5 mol%) | 1 | 3 |
| Seipin (S165A-S166A mutant) | CE (5 mol%) | 1 | 3 |

of the mixtures at different CE concentrations were then analyzed using ImageJ to measure the opacity.

### Imaging of lipid droplets and image analysis using polarized light

Cell experiments were imaged live at room temperature or 37 °C, when indicated, with an ×60 objective on a Zeiss LSM800 microscope. Images were then first analyzed using the segmentation "WEKA" plugin in FiJi. The algorithm was trained for each set of experiments and was error-checked by hand on test samples. WEKA gave a black-and-white segmentation of the images. Images were then treated with watershed and despeckle before using the "Analyze particles" plugin to determine the number and size of LDs.

### GUV and DEV preparation

Giant unilamellar vesicles (GUVs) were composed of 69 mol% DOPC, 30 mol% DOPE (Avanti polar lipids, Inc), 0.5 mol% Rhodamine-DOPE. For the nucleation experiments, CE/CE-bpy (18:2 TopFluor® cholesterol, Sigma-Aldrich) (99,5:0,5) GUVs were prepared by electroformation at 35 °C and imaged at room temperature. PLs and mixtures thereof in chloroform at 2.5 mM were dried on an indium tin oxide (ITO) coated glass plate. The lipid film was dessicated for 1 h. The chamber was sealed with another ITO-coated glass plate. The lipids were then rehydrated with a sucrose solution (275 ± 15 mOsm). Electroformation was performed using 100 Hz AC voltage at 1.4 Vrms and maintained for at least 2 h. This low voltage was used to avoid hydrolysis of water and dissolution of the titanium ions on the glass plate. GUVs were directly collected with a Pasteur pipette. CE artificial LDs (aLDs) were prepared as explained above. To make droplet-embedded vesicles (DEVs), GUVs were incubated with the aLDs for 10 min. The GUV-aLDs mixture was then placed on a glass coverslip pretreated with 10% (w/w) BSA and washed three times with buffer. Repeated experiments were conducted without treating the glass coverslip in order to burst the DEVs flat, and similar results were observed.

### FRAP experiments

FRAP experiments were performed on CE/CE-bpy-containing GUVs that had previously been deposited on untreated coverslips. The fluorescence recovery of CE-bpy signal is normalized by the fluorescence signal before bleaching. Images were acquired every second during the course of the experiment, with a Zeiss LSM 800 confocal microscope.

### Droplet/membrane partitioning analysis

The fluorescence signal of the CE-bodipy was quantified in the bilayer and in the droplet, by fitting a gaussian curve on a line profile in Fiji.

The ratio of the maxima in the droplet and in the bilayer is used as an output. In the case of burst DEVs, the fluorescence signal was averaged over a surface. The data have been normalized between the experiments in order to compare the droplet/membrane ratios between the different oils used.

### Thin layer chromatography

Cells were starved in 5% LPDS for 4 days + DGATi for the last 18 h, collected in 2% NaCl and lipids were extracted as described[64]. Equal amounts of each sample based on protein concentration were separated by thin layer chromatography (TLC) in a mixture of hexane: diethyl ether: acetic acid (80:20:1) as the running solvent. CE amounts were quantified from charred TLC plates using Fiji.

Cells were seeded at 70% confluence and allowed to incubate for 24 h under loading conditions. The next day, the cells were treated as indicated, and the cell culture media was removed. The cells were then washed three times with DPBS and detached using Trypsine-EDTA (Gibco™ #15400054). The cells were recovered, counted, and adjusted to have the same concentration in each sample. 200 μl of the cell suspension was mixed with 4 ml of CHCl₃/methanol (2:1) by vortexing, followed by 800 μl of water, also mixed by vortexing. The samples were allowed to sit for 1 h and vortexed again before separating the aqueous and organic phases. The aqueous phase was removed, and the organic phase was dried under a stream of nitrogen. The lipids were then resuspended in 50 μl of CHCl₃/methanol. Neutral lipids were separated using pre-coated silica-gel 60 plates and n-hexane/diethyl ether/glacial acetic acid (70:30:1) in an all-glass chromatography chamber. The plates were dried for 30 min in a fume hood and revealed by spraying with 50% aqueous sulfuric acid and heating in an oven at 115 °C for 30 min. The amounts of CE and TG were identified and quantified using standard compounds dissolved in CHCl₃/Methanol, and Fiji was used for quantification.

### Reporting summary

Further information on research design is available in the Nature Portfolio Reporting Summary linked to this article.

## Data availability

Data supporting the findings of this manuscript are available from the corresponding authors upon request. A reporting summary for this article is available as a Supplementary Information file. The datasets generated and analyzed during the current study are in the source data file or available from the corresponding authors on reasonable request. Source data are provided with this paper.

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

## Acknowledgements

We thank our teams for their critical discussions, Anna Uro and Bibi Hannikainen for excellent assistance with experiments, and Abel Szkalisty for help with image analysis. A.R.T. was supported by ANR-18-CE11-0012-01-MOBIL and ANR-21-CE11-0032-02-LIPRODYN, E.I. by the Academy of Finland grant 324929, Sigrid Juselius Foundation, Fondation Leducq and Jane and Aatos Erkko Foundation, L.V. by Orion research foundation, Paulo foundation, Aarne and Aili Turunen foundation, Alfred Kordelin foundation, Biomedicum Helsinki foundation, K. Albin Johansson foundation, Emil Aaltonen foundation, Maire Taponen foundation, Aarne Koskelo foundation, Waldemar von Frenckell foundation, The Finnish-Norwegian medical foundation, Paavo Ilmari Ahvenainen foundation, Finnish Medical Foundation, Paavo Nurmi foundation, The Doctoral Program in Biomedicine at the University of Helsinki. The I.V. lab is supported by the Helsinki Institute of Life Science (HiLIFE) Fellow program, Human Frontier Science Program (HFSP, project no. RGP0059/2019), Sigrid Juselius Foundation, and the Academy of Finland (project ID: 331349, 346135). We gratefully acknowledge CSC – IT Center for Science (Espoo, Finland) for providing ample computing resources and Biomedicum Imaging Unit supported by the Helsinki Institute of Life Science and Biocenter Finland for providing imaging resources.

## Author contributions

A.R.T., E.I., and I.V. designed the research. C.D. conducted all in vitro experiments and cell experiments using polarized light, with the help of M.C., M.O., and A.B. L.V conducted all cell experiments focusing on early LD formation, with the help of V.T.S. X.P. performed simulations, except for the energy calculation of seipin interaction with neutral lipids carried out by G.E. All authors analyzed the data. A.R.T and E.I. wrote the manuscript, reviewed and edited by all co-authors.

## Competing interests

The authors declare no competing interests.
