## [Peer Review File · Nature Communications]

Cholesterol esters form supercooled lipid droplets whose nucleation is facilitated by triacylglycerolsREVIEWER COMMENTS

Reviewer #1 (Remarks to the Author):

This mostly experimental paper examines the aggregation and phase behavior of cholesterol esters (CE) in a number of different systems related to lipid droplets (LD), including LD in several different cell types. Simulations are used to address question of aggregation of CE at low concentration in model bilayers, the effect of triglycerides (TG), and the role of the protein seipin. The central question addressed is how CEs in LDs are prevented from crystallizing, given that the melting temperature of pure CE is 44C, well above body temperature of 37C. This work is very timely, and would be of interest to a wide range of experimentalists, clinicians, and simulators of complex lipid assemblies. However, substantial revisions are required.

My main concern is that I don't believe that the authors provide adequate support that CL forms supercooled droplets in LDs, as stated in the title of the paper. Specifically, it is shown in the first paragraphs of page 5 that pure CL is a solid at 37C, and returns to a solid after it is melted and cooled. (An aside – isn't this common knowledge demanding a reference?). Mixing the CE solid with a buffer at 37C does not solubilize the CE. However, raising the temperature to 50C, adding buffer, vortexing, and cooling back to 37C leads to supercooled metastable lipid. This is interesting, but is not obviously related to what's going on in the cell, where there is no heating/cooling cycle, and where there are numerous other components.

Are the authors claiming that the CE in cellular lipid droplets should be viewed as supercooled? From what is written on page 7 line 171 it's a little confusing: "... above 20% CE, LDs would be in a supercooled state and above 90% CE in a liquid crystalline state." Does this mean that when CE is liquid crystalline it's not supercooled? It was stated that for the pure CE systems that the droplets are both supercooled and liquid crystalline.

Given the other components in cells (phospholipids, free fatty acids, free cholesterol) and the critical role of TG, I would be prone to think that the interior of an LD is an emulsion, or a complex mixture. For example, the liquid ordered phase in bilayers has large fractions of high melting lipids (DPPC or PSM), yet the Lo is relatively liquid-like because the low melting lipid and cholesterol break up long range order and subsequent freezing into a gel state. Is it possible that the LDs are more like that, as opposed to supercooled?

Page 5, line 109. It is stated that phospholipids cover the interface of the supercooled droplets with water. Does this mean that the PL form a monolayer around the CL, and essentially form a lipid droplet? Please clarify.

Page 6, first paragraph. It is stated that the CE in the supercooled droplets are mostly in a liquid crystal phase. Is this smectic or nematic? Later in the Discussion (page 14, line 356) it is stated that the CE forms radial lamellar phases, which I interpret to be smectic liquid crystal. It would be good not to leave things hanging.

It would be helpful to readers to list the composition of different lipid droplets, including free cholesterol and fatty acids.

The role of triglycerides is made clearly and convincingly in both the experiments and simulations, and the similarity of interaction of seipin with CE and TG is nicely explained. Some small questions:

Page 9, line 225. What is the temperature of the experiment?

Page 9, line 241. What is the lipid composition in the simulations. It's stated that it's similar to the ER, but that's not sufficiently specific. Why wasn't it the same as the experiment (7:3 DOPC:DOPE)?

Page 13, line 344. This last line was a little confusing. It was stated earlier that the interactions TG

and CE with seipin are comparable. Hence, I don't see that seipin has an enhanced ability to cluster TG over CE. Rather, the TG clusters form more readily and then associate with seipin to gain additional stability.

Minor editorial comments:

Page 2 (Abstract), line 26 (first sentence). I believe that the authors are stating the cholesterol is metabolized to cholesterol esters which are then stored in lipid droplets, but it's missing some words.

Page 4, line 85. Be more specific as to what is meant by "different membrane physical chemistry".

Page 7, line 159. Introduce the abbreviation OA for oleic acid, as OA is used afterward.

Reviewer #2 (Remarks to the Author):

Thiam et al present an interesting manuscript detailing the mechanisms by which cholesterol esters are nucleated into lipid droplets. Given the high melting temperature of cholesterol esters and propensity to form supercooled, metastable droplets at 37°C, the authors show that triacylglycerols are necessary to stabilize and catalyze the nucleation of cholesterol ester lipid droplets. The manuscript is well-written, and the results are laid out clearly. With the addition of appropriate statistics and proper controls, I think the manuscript could be of interest to the lipid droplet and biophysics community at large.

Major Concerns:

1. In Fig 2A-C, 1hr loading of cholesterol was used, whereas in 2D-E, 24hr loading was used. Can the authors explain the choice of incubation times and how these relate to the results? Also why was 200uM cholesterol used over 24hr instead of the 100uM in the previous experiment? What is the physical state of the increased number of LDs in panel 2C if not liquid crystalline? If TG synthesis is inhibited, how does 20uM OA increase the number of LDs over 10uM in the DGATi-treated cells (Fig 2C) (e.g. what is excess OA doing under these conditions)?
2. At early time points (2A with DGATi), does an increase in cholesterol in the cell trigger LD formation through increasing TG storage? Or in the early time points, could it be that the Bodipy is staining the lysosomes or other intracellular lipid pools in the case of extreme excess of cholesterol loading (2C) prior to incorporation into CE or TAGs? Controls to rule this out need to be included.
3. I find it interesting that there are no LC LDs in Cos7 or HeLa cells (Fig S2C-F). Why did the authors include this data, but not discuss it further? The reason for showing different cell types is to show generality (or lack thereof); thus, it would be beneficial to explain these results more. Because these results come in the section of the manuscript that is titled "Inhibition of TG synthesis compromises and stimulation of TG synthesis enhances CE LD formation in cells" it is odd that there is no mention of CE LDs in these other cell types. Additionally, although 200uM cholesterol was fed to these cells, there is no data to show that the LDs are indeed CE-containing. The polarized light experiment for the Cos7 seems unnecessary if the authors are not going to discuss the lack of LC LDs in these cells under the cholesterol-loading experiments performed. More discussion of why LC LDs don't exist in these cells is warranted.
4. On line 221, it is assumed that the excess cholesterol fed to the cells is all being funneled into CE and that the reason that LC LDs are not forming without TG is because CE doesn't efficiently condense in the cellular conditions. The atomistic modeling suggests this could be true, but the authors need to show that the CE content of cells is indeed higher in their feeding conditions and what the magnitude of change is to be able to relate it to the more simple in vitro and in silico models.

5. The statistical analyses of all experiments in this manuscript are missing or not appropriately performed. For example, counting >1000 LDs over two experiments and combining the results does not inform the reader (or experimenter) of the variability across experiments (where significant variability might occur due to differences in cell states and/or FA- or cholesterol-feeding conditions). Proper reporting of the mean and SD of independent experiments with statistical analysis performed across >3 independent experiments needs to be done and shown.

Minor Points of Concern:

1. Fig 1G, what are the arrows pointing to?

2. Fig 2F has misspelling in Fraction.

3. Lines 182 and 185 have very similar sentences that might do better to be restated more differently from each other.

4. The figure legends need to be more clear when maximum projections are shown versus one confocal plane.

5. The sentence on line 361 "Such a situation could be deleterious to the ER given that the concentration of membrane lipids such as cholesterol, PA, and diacylglycerols are kept at low levels (39)." is unclear to me.

RESPONSE TO REVIEWERS' COMMENTS

Reviewer #1 (Remarks to the Author):

This mostly experimental paper examines the aggregation and phase behavior of cholesterol esters (CE) in a number of different systems related to lipid droplets (LD), including LD in several different cell types. Simulations are used to address question of aggregation of CE at low concentration in model bilayers, the effect of triglycerides (TG), and the role of the protein seipin. The central question addressed is how CEs in LDs are prevented from crystallizing, given that the melting temperature of pure CE is 44C, well above body temperature of 37C. This work is very timely, and would be of interest to a wide range of experimentalists, clinicians, and simulators of complex lipid assemblies. However, substantial revisions are required.

We are grateful to this reviewer for raising important points that have helped to improve our manuscript. We have put a lot of effort in explaining and introducing all the concepts. We hope our answers are satisfactory.

My main concern is that I don't believe that the authors provide adequate support that CL forms supercooled droplets in LDs, as stated in the title of the paper. Specifically, it is shown in the first paragraphs of page 5 that pure CL is a solid at 37C, and returns to a solid after it is melted and cooled. (An aside – isn't this common knowledge demanding a reference?).

We have now added the following reference that studied the different phases of cholesterol oleate (isotropic, cholesteric nematic and cholesteric smectic).
(<https://www.tandfonline.com/doi/abs/10.1080/00268948108073588?journalCode=gmc16>)

We assume that the referee means here CE (cholesterol ester) and not CL (cholesterol).

What we indicated in the cited paragraph (page 5) was that CE is solid at 37°C in bulk. In droplets, it can stay liquid. This indicates that having CE in a liquid droplet state at 37°C, i.e. temperature at which it should be solid, corresponds to a metastable liquid state, commonly known as supercooling (e.g. see "Supercooling of liquids." Proceedings of the Royal Society of London. Series A. Mathematical and Physical Sciences 215.1120 (1952): 43-46.).

Mixing the CE solid with a buffer at 37C does not solubilize the CE. However, raising the temperature to 50C, adding buffer, vortexing, and cooling back to 37C leads to supercooled metastable lipid. This is interesting, but is not obviously related to what's going on the cell, where there is no heating/cooling cycle, and where there are numerous other components.

We are very thankful that the reviewer raised this point that needs to be made clearer as it is a key point of our manuscript. Therefore, we have developed this discussion further in the manuscript.

Briefly, the answer to this question is purely thermodynamic, the phase change of pure chemical species being driven by the Gibbs Free Energy change:

$$\Delta G = \Delta H - T\Delta S$$

It is important to consider that, as the equation is based on state functions, the chosen thermodynamic path should not be important for the reaction. Here we chose to navigate through phase changes using temperature as the variable but one could also decide to use pressure or chemical potential, for instance. Indeed, the development of the differential Gibbs Free Energy equation simply gives:

$$dG = -SdT + VdP + \mu_i dn_i$$

Regardless of the free parameter tuning the system, the important point is that the value of the free energy of the liquid defines the physical state. In our case, we found that the molecules could be trapped in a Liquid Crystalline metastable phase in vitro (figure 1) and in cellulo (figure 2).

It is true that cells will not shift temperature or pressure to make supercooled LDs but can tune chemical potentials instead. Indeed, in the cell context, CE is synthesized and delivered to the hydrophobic region of the ER membrane. At a critical concentration relative to phospholipids, CE molecules transit to droplets (seen in vitro and in cellulo). Such transition occurs due to the lower chemical potential when partitioning into droplets than when being free, i.e. in contact with phospholipids (Thiam & Foret 2016, BBA); it is not provoked by temperature shifts. This process was mimicked in vitro and in silico in figure 3. In the latter figure, temperature was not changed, and we demonstrated that, as in cells, CE droplets can be formed by reaching a critical concentration in the membrane (variations are again via chemical potential and not temperature despite that

they both contribute to the Gibbs free energy). The made LDs were indeed supercooled as seen via their liquid crystalline structure under polarized light excitation.

Are the authors claiming that the CE in cellular lipid droplets should be viewed as supercooled? From what is written on page 7 line 171 it's a little confusing: "... above 20% CE, LDs would be in a supercooled state and above 90% CE in a liquid crystalline state." Does this mean that when CE is liquid crystalline it's not supercooled? It was stated that for the pure CE systems that the droplets are both supercooled and liquid crystalline.

We thank the reviewer for pointing this. We should have been more explicit on this particular point. Supercooling takes place when a substance is cooled below its freezing point without becoming solid. On one hand, in bulk (i.e. in the test tube with no aqueous phase) and at 37°C, when the concentration of CE in TG is above 20%, the mixture exhibits a phase separation: a solid, mostly CE phase, and, a liquid, mostly TG phase. On the other hand, in droplets and at 37°C, the mixture remained liquid for any CE concentration and even at 100% of CE. This showed that confining CE in droplets prevents CE from reaching the stable solid state, defining it as supercooling.

Above 20 % CE, the droplets are supercooled. The liquid crystalline phase is only a particular state of supercooling that gives information on how concentrated and organized the CE in the droplets is.

Given the other components in cells (phospholipids, free fatty acids, free cholesterol) and the critical role of TG, I would be prone to think that the interior of an LD is an emulsion, or a complex mixture. For example, the liquid ordered phase in bilayers has large fractions of high melting lipids (DPPC or PSM), yet the L_0 is relatively liquid-like because the low melting lipid and cholesterol break up long range order and subsequent freezing into a gel state. Is it possible that the LDs are more like that, as opposed to supercooled?

Excess of lipids (e.g. cholesterol, oleic acid) is deleterious for the cells. Hence, they are used or stored as two types of lipids, surface and bulk lipids. Neutral lipids, such as TG and CE, are the major bulk lipids and are stored in the lipid droplet core. Surface lipids (e.g. phospholipids) are the main components of cellular membranes. Only bulk lipids have been reported to be in the lipid droplet core (from different lipidomics studies of LDs), ruling out the case of LDs' interiors as being a mixture (Tauchi-Sato, Kumi, et al. "The surface of lipid droplets is a phospholipid monolayer with a unique fatty acid composition." *Journal of Biological Chemistry* 277.46 (2002): 44507-44512.)

In vitro, where all components are controlled, the birefringent Maltese crosses are obtained only above 85% CE relative to TG, at 30°C. In cells, the appearance of such features when cells were exclusively fed with cholesterol suggests that the cellular LDs had likewise a high content of CE. We have now done thin layer chromatography analysis of the cells and found that the LDs contained CE, which better accounts for our statement.

Page 5, line 109. It is stated that phospholipids cover the interface of the supercooled droplets with water. Does this mean that the PL form a monolayer around the CL, and essentially form a lipid droplet? Please clarify.

Yes, even the supercooled liquid crystalline LDs are covered by a monolayer of phospholipids, which act as surfactants. Please note again that we made CE (cholesterol ester) and not CL(cholesterol) LDs.

Page 6, first paragraph. It is stated that the CE in the supercooled droplets are mostly in a liquid crystal phase. Is this smectic or nematic? Later in the Discussion (page 14, line 356) it is stated that the CE forms radial lamellar phases, which I interpret to be smectic liquid crystal. It would be good not to leave things hanging.

We thank the referee for highlighting this point. We now precise in the text the nature of the phase, we apologize for leaving this point unclear: the liquid crystalline phase is a smectic phase.
https://www.researchgate.net/figure/Molecular-structure-of-a-cholesteryl-oleate-b-cholesteryl-oleyl-carbonate-in-each_fig2_223401854

It would be helpful to readers to list the composition of different lipid droplets, including free cholesterol and fatty acids.

So far, there is no free cholesterol nor free fatty acids reported in lipid droplets. The most abundant lipids in the droplets are neutral lipids (triglycerides and cholesterol esters) and phospholipids such as PC, PE, and PI. This was already described in the early 2000s by (Tsuchi-Sato, Kumi, et al. "The surface of lipid droplets is a phospholipid monolayer with a unique fatty acid composition." *Journal of Biological Chemistry* 277.46 (2002): 44507-44512.) and by Thiam & Ikonen 2021, *TCB* (10.1016/j.tcb.2020.11.006).

The role of triglycerides is made clearly and convincingly in both the experiments and simulations, and the similarity of interaction of seipin with CE and TG is nicely explained. Some small questions:

Page 9, line 225. What is the temperature of the experiment?

GUVs were made at 35°C and imaged at room temperature. This information is added to the methods section.

Page 9, line 241. What is the lipid composition in the simulations. It's stated that it's similar to the ER, but that's not sufficiently specific. Why wasn't it the same as the experiment (7:3 DOPC:DOPE)?

The lipid composition of the membranes used in our simulations is the same as in our previous work (Prasanna et al., *PLoS Biol* 19, e3000998 (2021)). In these simulation models, we have ensured that the lipid composition of the simulation model matches the realistic composition as accurately as possible. We have now clarified this point in the revised manuscript.

However, we agree that the simulation data presented in the article should have included also the results matching lipid concentrations used in our experiments. For completeness, we here present the simulation data performed at varying CE concentrations (5 mol%, 10 mol%, and 15 mol% CE) in a DOPC/DOPE (70/30) bilayer, thus corresponding to conditions used in experiments. The new results (Figure S3D) are consistent with the data given in the original manuscript (Figure 3F,G). When the new results presented in Figure S3D are compared with the data in Figure 3F,G (main manuscript), one finds in both cases that CE forms nanoscale clusters when the CE concentration reaches ~10 mol%. Also, in both cases, for CE concentration of 10 mol%, the fraction of clustered CE is quantitatively identical. We conclude that when simulations are conducted with lipid compositions matching experimental conditions, CE forms stable clusters at higher CE concentrations, as stated in the original manuscript.

.Page 13, line 344. This last line was a little confusing. It was stated earlier that the interactions TG and CE with seipin are comparable. Hence, I don't see that seipin has an enhanced ability to cluster TG over CE. Rather, the TG clusters form more readily and then associate with seipin to gain additional stability.

We agree with this point and that is the message we intended to convey. We have now tried to rephrase this to make it clearer.

Minor editorial comments:

Page 2 (Abstract), line 26 (first sentence). I believe that the authors are stating the cholesterol is metabolized to cholesterol esters which are then stored in lipid droplets, but it's missing some words.
This has been corrected.

Page 4, line 85. Be more specific as to what is meant by "different membrane physical chemistry".
This has been corrected.

Page 7, line 159. Introduce the abbreviation OA for oleic acid, as OA is used afterward.
This has been corrected.

Reviewer #2 (Remarks to the Author):

Thiam et al present an interesting manuscript detailing the mechanisms by which cholesterol esters are nucleated into lipid droplets. Given the high melting temperature of cholesterol esters and propensity to form supercooled, metastable droplets at 37°C, the authors show that triacylglycerols are necessary to stabilize and catalyze the nucleation of cholesterol ester lipid droplets. The manuscript is well-written, and the results are laid out clearly. With the addition of appropriate statistics and proper controls, I think the manuscript could be of interest to the lipid droplet and biophysics community at large.

We thank this reviewer for having examined our manuscript and raising interesting points to elaborate on. We hopefully have satisfactorily addressed her/his questions with additional experiments and analyses.

Major Concerns:

1. In Fig 2A-C, 1hr loading of cholesterol was used, whereas in 2D-E, 24hr loading was used. Can the authors explain the choice of incubation times and how these relate to the results?

At 1 h, the major information that was extracted was the LD number, whose tendency matched the one at 24 h. However, because LDs are tiny at such a time point, the polarized light analysis could not allow to pinpoint their internal structure and content.

At 24 h, LDs are big enough to be characterized by polarized light.

Combining these two analyses gave us a rather exhaustive knowledge on both the number of LD assembled and their neutral lipid content and organization.

Also why was 200uM cholesterol used over 24hr instead of the 100uM in the previous experiment?

100uM loadings were used along with shorter timescales (30 or 60 min) in order to count early nucleated droplets. However, polarized light signal cannot be resolved for such small droplets (panel 2C). Thus, we used 200uM loadings over 24 h to be able to observe polarized light signal from bigger droplets.

What is the physical state of the increased number of LDs in panel 2C if not liquid crystalline?

With our polarized technique, we can only assume that the LDs in 2C were under liquid crystalline state. Indeed, at 1 h, the LDs are tiny.

If TG synthesis is inhibited, how does 20uM OA increase the number of LDs over 10uM in the DGATi-treated cells (Fig 2C) (e.g. what is excess OA doing under these conditions)?

When cells are loaded with 10 μ M or 20 μ M OA *together with 100 μ M cholesterol* in the TG synthesis inhibition conditions, more CEs are formed with 20 μ M OA than with 10 μ M OA, as shown in revised Figure S2E. Thus, under these conditions the excess OA is esterified to cholesterol.

2. At early time points (2A with DGATi), does an increase in cholesterol in the cell trigger LD formation through increasing TG storage? Or in the early time points, could it be that the Bodipy is staining the lysosomes or other intracellular lipid pools in the case of extreme excess of cholesterol loading (2C) prior to incorporation into CE or TAGs? Controls to rule this out need to be included.

In Fig. 2A at early time points an increase in cholesterol leads to an increased generation of CEs and thereby the formation of CE LDs. We do not exclude the possibility that some TGs are synthesized in these cells despite the presence of DGAT inhibitors, but at early time points (1 h) of cholesterol loading, this is negligible compared to the synthesis of CEs as the increase in CEs is evident while TGs were not detectable biochemically. We have now included the measurement of CEs under the early 1 h time point in Fig S2A.

The reviewer also questioned if some of the LD dye (we used LD540, not Bodipy) would be staining the lysosomes or other intracellular lipid pools at the early time points.

Indeed, lysosomes would be the most likely compartment to be picked up by lipophilic dyes and morphologically staining similarly to LDs. We have therefore performed double labeling with a lysosomal tracer (dextran) and LD540 under the early 1 h time point. This shows that LD540 is mostly not colocalizing with dextran and argues that the dye is not prominently staining lysosomes under these conditions. These data have been added to Fig S2B.

3. I find it interesting that there are no LC LDs in Cos7 or HeLa cells (Fig S2C-F). Why did the authors include this data, but not discuss it further? The reason for showing different cell types is to show generality (or lack thereof); thus, it would be beneficial to explain these results more. Because these results come in the section of the manuscript that is titled "Inhibition of TG synthesis compromises and stimulation of TG synthesis enhances CE LD formation in cells" it is odd that there is no mention of CE LDs in these other cell types. Additionally, although 200uM cholesterol was fed to these cells, there is no data to show that the LDs are indeed CE-containing. The polarized light experiment for the Cos7 seems unnecessary if the authors are not going to discuss the lack of LC LDs in these cells under the cholesterol-loading experiments performed. More discussion of why LC LDs don't exist in these cells is warranted.

We thank the reviewer for bringing up this point.

We did not include polarized light signal data in Fig S2F because we only wanted to show the similar trend when DGATs were inhibited in HeLa cells. The polarized light signal from the HeLa cells experiment has now been added in complement of Fig S2F. It shows that LC LDs can also be found in HeLa cells.

The data in Cos7 cells were likewise to show that TG synthesis is mandatory for packaging CE into LDs. Indeed, blocking TG synthesis by inhibiting DGATs almost completely inhibited LD formation upon cholesterol loading in this cell line (not in A431 nor HeLa cells, nor macrophages or Huh7 liver cells, which were not shown). Yet, we were struck as the referee that in this cell line no liquid crystalline phase was observed when feeding with cholesterol. We interpreted this to suggest that this particular cell line, for whatever metabolic reasons, needs/makes more TG to handle CE accumulation.

Now we have added thin layer chromatography data showing that both in A431 and Cos7 cells, CE accumulated when the cells were fed with cholesterol for 24hr (longer time points).

Therefore, Cos7 cells made CE and require TG to package them in LDs. Again, the point of the Cos7 data were more to demonstrate that TG was required for CE containing LD assembly, which was a striking result in this cell line with a higher TG/CE ratio.

In our revised version, we elaborate succinctly on this point, i.e. on the sensitivity of CE LD formation to DGAT inhibition and the absence of LC phases in LDs of these cells.

4. On line 221, it is assumed that the excess cholesterol fed to the cells is all being funneled into CE and that the reason that LC LDs are not forming without TG is because CE doesn't efficiently condense in the cellular conditions. The atomistic modeling suggests this could be true, but the authors need to show that the CE content of cells is indeed higher in their feeding conditions and what the magnitude of change is to be able to relate it to the more simple in vitro and in silico models.

Thank you for bringing up this point. As requested, we have now added data to demonstrate that the cellular CE content is indeed higher in the cholesterol feeding condition (Fig S2A).

5. The statistical analyses of all experiments in this manuscript are missing or not appropriately performed. For example, counting >1000 LDs over two experiments and combining the results does not inform the reader (or experimenter) of the variability across experiments (where significant variability might occur due to differences in cell states and/or FA- or cholesterol-feeding conditions). Proper reporting of the mean and SD of independent experiments with statistical analysis performed across >3 independent experiments needs to be done and shown.

There is indeed some variability between experiments, despite our conditions where the seeding of cells to experiments, their cultivation and loading conditions are carefully controlled. Moreover, LD numbers vary physiologically between cells much more than for instance lysosome numbers. We have increased the number of experiments and cells analyzed and improved the reporting on statistical analyses. For the high-content imaging experiments with hundreds of cells automatically analyzed per condition, similar reporting as in previous publications was employed.

Minor Points of Concern:

1. Fig 1G, what are the arrows pointing to?

The arrows point to LDs and their respective polarized light signal.

2. Fig 2F has misspelling in Fraction.

This has now been corrected.

3. Lines 182 and 185 have very similar sentences that might do better to be restated more differently from each other.

This has now been rephrased.

4. The figure legends need to be more clear when maximum projections are shown versus one confocal plane.

This is now detailed.

5. The sentence on line 361 "Such a situation could be deleterious to the ER given that the concentration of membrane lipids such as cholesterol, PA, and diacylglycerols are kept at low levels (39)." is unclear to me.

We apologize for the unclear phrasing. We thereby wanted to explain that as for cholesterol, PA, and diacylglycerols, high levels of CE could be deleterious for the membrane and that it needed to be packaged into lipid droplets.

REVIEWERS' COMMENTS

Reviewer #1 (Remarks to the Author):

The authors have satisfactorily responded to all of my comments. Publish

Reviewer #2 (Remarks to the Author):

In general, Dumesnil et al have adequately answered the majority of my critiques. The manuscript is clear and well-presented. I do still suggest that better statistical methods be used and shown. Combining 500+ LDs across multiple experiments is not the best way of presenting this data and can be viewed as skewing the results. The variability across experiments is equally important as the results/trends of the data in the individual experiments. As such, combining these data points into one bar graph is not acceptable. Rather, a scatter plot showing the means from the independent experiments with standard deviation between these could be shown. Best would be showing the means from all experiments (and doing statistical tests from this) AND showing the spread through a scatter plot of all points from a representative experiment, either in the supplement or as an inset. As it is now, the manuscript has zero statistics reported which is unacceptable at any level of publishing.

Additionally, some crucial details of the methods are still missing. Prominently, which "biochemical analysis" was performed to quantitate ng cholesterol esters / ug protein? Is this a TLC somehow, or was an Amplex Red cholesterol assay used to obtain this level of accuracy? Panel S2E suggest this is the same biochemical assay but the results presented appear to be the quantitation of the TLC (which should be more explicitly stated as only being SEMI-quantitative in nature).

RESPONSE TO REVIEWERS' COMMENTS

In general, Dumesnil et al have adequately answered the majority of my critiques. The manuscript is clear and well-presented. I do still suggest that better statistical methods be used and shown. Combining 500+ LDs across multiple experiments is not the best way of presenting this data and can be viewed as skewing the results. The variability across experiments is equally important as the results/trends of the data in the individual experiments. As such, combining these data points into one bar graph is not acceptable. Rather, a scatter plot showing the means from the independent experiments with standard deviation between these could be shown. Best would be showing the means from all experiments (and doing statistical tests from this) AND showing the spread through a scatter plot of all points from a representative experiment, either in the supplement or as an inset. As it is now, the manuscript has zero statistics reported which is unacceptable at any level of publishing.

We thank the referee for these critical remarks which we all address now. All data points are displayed, individually and collapsed. Statistical tests are done for all.

Additionally, some crucial details of the methods are still missing. Prominently, which "biochemical analysis" was performed to quantitate ng cholesterol esters / ug protein? Is this a TLC somehow, or was an Amplex Red cholesterol assay used to obtain this level of accuracy? Panel S2E suggest this is the same biochemical assay but the results presented appear to be the quantitation of the TLC (which should be more explicitly stated as only being SEMI-quantitative in nature).

TLC was used to determine lipids.